# Species-specific host factors rather than virus-intrinsic virulence determine primate lentiviral pathogenicity

Simone Joas[1], Erica H. Parrish[2,9], Clement W. Gnanadurai[1,10], Edina Lump[1], Christina M. Stürzel[1], Nicholas F. Parrish[2,11], Gerald H. Learn[2], Ulrike Sauermann[3], Berit Neumann[3], Kerstin Mätz Rensing[3], Dietmar Fuchs[4], James M. Billingsley[5], Steven E. Bosinger[5], Guido Silvestri[5], Cristian Apetrei[6], Nicolas Huot[7,8], Thalia Garcia-Tellez[7], Michaela Müller-Trutwin[7], Dominik Hotter[1], Daniel Sauter[1], Christiane Stahl-Hennig[3], Beatrice H. Hahn[2] & Frank Kirchhoff[1]

HIV-1 causes chronic inflammation and AIDS in humans, whereas related simian immuno-deficiency viruses (SIVs) replicate efficiently in their natural hosts without causing disease. It is currently unknown to what extent virus-specific properties are responsible for these different clinical outcomes. Here, we incorporate two putative HIV-1 virulence determinants, i.e., a Vpu protein that antagonizes tetherin and blocks NF-κB activation and a Nef protein that fails to suppress T cell activation via downmodulation of CD3, into a non-pathogenic SIVagm strain and test their impact on viral replication and pathogenicity in African green monkeys. Despite sustained high-level viremia over more than 4 years, moderately increased immune activation and transcriptional signatures of inflammation, the HIV-1-like SIVagm does not cause immunodeficiency or any other disease. These data indicate that species-specific host factors rather than intrinsic viral virulence factors determine the pathogenicity of primate lentiviruses.

[1] Institute of Molecular Virology, Ulm University Medical Center, 89081 Ulm, Germany. [2] Departments of Medicine and Microbiology, University of Pennsylvania, Philadelphia, PA 19104, USA. [3] German Primate Centre, 37077 Göttingen, Germany. [4] Division of Biological Chemistry, Biocenter Innsbruck Medical University, Center for Chemistry and Biomedicine, A-6020 Innsbruck, Austria. [5] Emory Vaccine Center and Yerkes National Primate Research Center, Emory University, Atlanta, GA 30322, USA. [6] WA Center for Vaccine Research, University of Pittsburgh, Pittsburgh, PA 15261, USA. [7] Institut Pasteur, Unité HIV, Inflammation and Persistence, Paris 75015, France. [8] Vaccine Research Institute, Hôpital Henri Mondor, Créteil 94010, France. [9] Present address: Department of Pathology, Microbiology and Immunology, Vanderbilt University Medical Center, Nashville, TN 372327, USA. [10] Present address: Department of Veterinary Pathology, University of Georgia, Athens, GA 30602, USA. [11] Present address: Department of Surgery, Vanderbilt University Medical Center, Nashville, TN 37232, USA. Correspondence and requests for materials should be addressed to F.K. (email: frank.kirchhoff@uni-ulm.de)

Since the beginning of the AIDS pandemic, HIV-1 has caused over 30 million deaths. In contrast, many related primate lentiviruses replicate to high levels in their natural primate hosts without causing disease[1,2]. One difference between HIV-1 and these viruses, which have been misleadingly designated simian immunodeficiency viruses or SIVs, is that only the former is associated with high levels of chronic inflammation and CD4+ T cell loss, which correlate with disease progression[3,4]. It is a long-standing question whether these strikingly different outcomes of primate lentiviral infections are primarily determined by viral properties or host factors, or a combination of both[5,6].

Comprehensive analysis of two of over 40 naturally SIV-infected African primates (sooty mangabeys and African green monkeys) have provided clear evidence that host factors play an important role in determining viral pathogenicity. Both SIVsmm and SIVagm are nonpathogenic in their natural hosts, but the same viruses are virulent in experimentally infected macaques[7–9]. It has been proposed that limited expression of the entry coreceptor CCR5 in sooty mangabeys might protect critical CD4+ T cell subsets from SIVsmm infection and depletion[10]. Similarly, CD4 downmodulation may render essential central memory CD4+ T cell subsets resistant to SIVagm infection in African green monkeys[11]. Furthermore, recent analysis of the entire sooty mangabey genome revealed mutations in several immune genes that might contribute to the lack of harmful chronic immune activation upon SIVsmm infection[12].

Well-adapted natural primate hosts appear to have evolved specific mechanisms that protect them from the pathogenicity of their own SIV strains. However, HIV-1 and its closest SIV relatives share unique features that are absent from non-pathogenic primate lentiviruses such as SIVsmm and SIVagm and might be associated with increased virulence[6]. Specifically, the simian precursors of HIV-1 acquired an additional accessory gene encoding the viral protein U (Vpu) that allows pandemic HIV-1 M strains to counteract the restriction factor tetherin[13,14] and to inhibit innate immune responses by suppressing NF-κB activity[15]. Furthermore, this vpu gene acquisition coincided with a loss of the CD3 downmodulation activity of the viral protein Nef, which plays an important role in primate lentiviral replication and pathogenesis[16–19]. The CD3 molecule is a key component of the T cell receptor (TCR) complex and expressed on the surface of all T cells. CD3 is critical for the transmission of intracellular signals upon TCR-CD3 binding to peptide-MHC complexes on Antigen-presenting cells (APCs) and hence the activation of T cells in response to foreign antigens. Consequently, the Nef proteins of HIV-1 and its closest simian relatives heightens the responsiveness of infected T cells to TCR-CD3 mediated stimulation, while most primate lentiviral Nefs block this activation and disrupt the immunological synapse between infected T cells and APCs[18–20]. The lack of Nef-mediated downmodulation of CD3 is associated with increased levels of immune activation and apoptosis in virally infected CD4+ T cell cultures[21]. In addition, expression of a Nef protein that contains an immunoreceptor tyrosine-based activation motif (ITAM) and induces strong T lymphocyte activation is associated with acute pathogenicity in SIV-infected rhesus macaques[22]. Finally, SIVcpz, the direct precursor of pandemic HIV-1 strains, causes AIDS in wild chimpanzees[23]. These findings led to the hypothesis that the combination of a Vpu protein with a Nef protein that lacks the ability to suppress T cell activation by CD3 downmodulation generated a uniquely pathogenic virus capable of causing harmful chronic immune activation and disease progression.

To directly test whether these HIV-1-specific virulence determinants would be sufficient to increase the pathogenic potential of a non-pathogenic primate lentivirus in a natural host model, we generated infectious molecular clones of SIVagm expressing HIV-1-like Vpu and Nef proteins individually and in combination. We found that all SIVagm chimeras replicated efficiently in vitro. However, only the SIVagm construct that expressed both the HIV-1-like Vpu and Nef protein maintained sustained high viral loads in vivo. Although monkeys infected with this virus exhibited increased levels of chronic immune activation, the HIV-1-like SIVagm construct failed to cause accelerated CD4+ T cell depletion and immunodeficiency. Thus, despite maintenance of the HIV-1-specific accessory protein functions over almost 5 years of follow-up and increased inflammatory gene expression, these viral properties did not supersede the host factors that protect African green monkeys against lentiviral pathogenicity.

## Results

**SIVagm constructs with HIV-1-like Vpu and Nef functions**. To generate SIVagm constructs with properties that are characteristic for HIV-1, we introduced the SIVgsn71 vpu and/or the HIV-1 NA7 nef allele into a wild-type (WT) SIVagm transmitted/ founder (TF) molecular clone (Fig. 1a)[24]. We selected the SIVgsn rather than an HIV-1 vpu gene, because tetherin is counteracted in a species-specific manner and only the former is capable of antagonizing African green monkey (AGM) tetherin (Fig. 1b). Functional studies confirmed that the SIVgsn Vpu reduced cell surface expression of AGM tetherin (BST-2) as well as human CD4, NTB-A, and CD1d (Fig. 1c and Supplementary Fig. 1a) and suppressed activation of NF-κB (Fig. 1d). We tested the AGM ortholog of tetherin because tetherin is frequently counteracted in a species-dependent manner[25–27]. However, since Nef and Vpu-mediated counteraction of immune receptors is usually species-independent and since the residues necessary for modulation are highly conserved, we tested the human versions of CD4, NTB-A, and CD1d[27–31]. Our finding that the SIVgsn Vpu downmodulates human CD4, NTB-A, and CD1d molecules is in agreement with these previous data.

The NA7 nef allele was selected because it has been well-characterized and proven highly active in previous studies[32,33]. Similar to SIVagm Nef, the HIV-1 Nef downmodulated CD4, MHC-I, CD28, and (less potently) CXCR4 from the surface of infected human CD4+ T cells (Fig. 1e and Supplementary Fig. 1b). Furthermore, both SIVagm and HIV-1 Nef enhanced surface expression of the MHC-II-associated invariant chain (Ii, CD74) in the monocyte-derived THP-1 cell line and in human PBMCs (Supplementary Figs. 1b and 1c). However, the HIV-1 Nef was inactive in modulating CD3 (Fig. 1e and Supplementary Fig. 1b). It has been reported that the NA7 Nef shows activity against sooty mangabey but not human or macaque tetherin[26]. We found that the NA7 Nef also failed to counteract AGM tetherin, which is closely related to the rhesus orthologue (Fig. 1f). Altogether, these functional analyses showed that the SIVgsn71 vpu and HIV-1 NA7 nef genes recapitulated accessory gene functions that are characteristic for HIV-1.

To generate chimeric constructs, we introduced the SIVgsn vpu allele upstream of the SIVagm env gene mimicking its position in HIV-1 (Fig. 1a). To replace the SIVagm nef gene, we mutated the original nef initiation codon and inserted the HIV-1 nef downstream of the env gene. Subsequently, an SIVagm construct containing both the SIVgsn vpu and the HIV-1 nef was generated. The SIVagm chimeras were designated GU (SIVgsn Vpu), 1N (HIV-1 Nef) and GU1N (SIVgsn Vpu and HIV-1 Nef). The GU and GU1N constructs contained a strong Kozak sequence (AGT) upstream of the vpu gene, which inhibited the expression of the downstream env gene. Mutation of AGT to CGT enabled leaky scanning and increased envelope glycoprotein (Env) expression (Supplementary Fig. 1d) and infectious virus production (Supplementary Fig. 1e). Thus, we used GU and GU1N

constructs containing the CGT *vpu* Kozak sequence in all subsequent experiments.

**Chimeric SIVagm constructs express functional Vpu and Nef.** To determine the replication capacity of the SIVagm constructs, we performed in vitro infection experiments with virus stocks normalized for p27 capsid content. The three chimeric SIVagm constructs infected TZM-bl cells (Fig. 1g) and replicated with similar, albeit moderately delayed, kinetics compared to the WT virus in MOLT-4 clone eight cells and AGM PBMCs (Fig. 1h). As expected, the WT SIVagm clone and its *vpu* containing GU and GU1N derivatives produced more infectious virus in the presence of AGM tetherin, whereas SIVagm constructs containing a disrupted *nef* gene (*nef**) or the HIV-1 Nef alone (1N) were unable to counteract tetherin (Fig. 1i). Infection of AGM PBMCs confirmed that viral constructs expressing the parental SIVagm Nef (WT, GU) and the HIV-1 Nef (1N, GU1N) differed markedly in their CD3 downmodulation function (Fig. 1j and Supplementary Fig. 1f), while all SIVagm constructs were capable of downmodulating MHC-I and CD28. AGM PBMCs infected with SIVagm GU (expressing two tetherin antagonists) exhibited the lowest and those infected with the 1N (lacking a tetherin antagonist) the highest tetherin surface expression levels (Fig. 1j). Vpu amplified the effects of HIV-1 Nef on MHC-I and CD28, possibly by modulating HLA-C[34] and/or by suppressing their expression through inhibition of the activity of the transcription

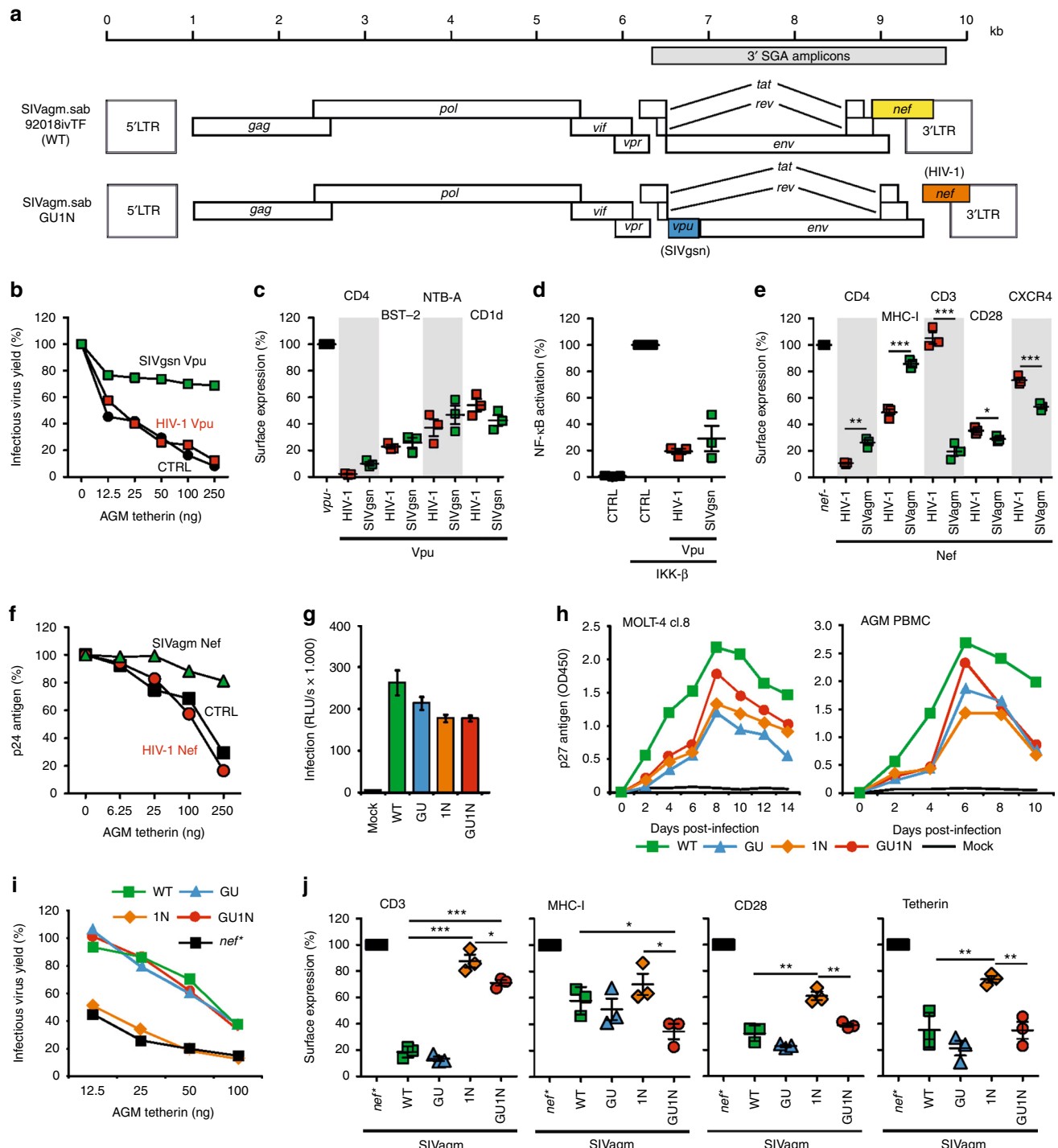

factor NF-κB[15]. Thus, all chimeric SIVagm constructs (GU, 1N, GU1N) replicated efficiently in vitro and expressed Vpu and Nef proteins that recapitulated HIV-1 accessory protein functions.

**Matched Vpu and Nef are required for SIV replication in vivo.** To examine the effects of HIV-1-like Vpu and Nef proteins on SIVagm replication in vivo, we infected 12 AGMs, inoculating groups of three animals with each of the four SIVagm constructs. In contrast to the in vitro results, in vivo replication efficiencies differed considerably (Fig. 2a and Supplementary Fig. 2a). Animals infected with WT SIVagm exhibited mean peak plasma viral loads of $1.7 \times 10^6$ copies/ml and set point viral loads of ~50,000 vRNA copies/ml. In contrast, AGMs inoculated with the SIVagm GU construct had no detectable viremia. The exception was AGM14627, which became viremic 223 weeks post-infection (wpi) (Supplementary Fig. 2a) due to an accidental infection between cage neighbors (see 'Maintenance of intact *vpu* and *nef* genes'). The SIVagm 1N construct replicated to high vRNA titers during acute infection (mean peak viral load $6.6 \times 10^5$ copies/ml) in all three animals but viremia dropped ~200-fold six wpi in two animals (AGM14626 and AGM14630) and became undetectable in one (AGM14631). All animals infected with this virus also exhibited a rapid decline in cell-associated infectious virus following an initial peak at two wpi, suggesting effective virus control (Supplementary Fig. 2b). The HIV-1-like double chimera SIVagm GU1N showed ~20-fold lower peak vRNA loads ($8.5 \times 10^4$ copies/ml) compared to WT SIVagm during acute infection, while the average set-point viral load from 16 to 55 wpi was only ~5-fold reduced. Interestingly, the mean vRNA loads subsequently increased and were as high as in WT SIVagm infected AGMs in the third (WT: $1.9 \times 10^5$ vs. GU1N $1.8 \times 10^5$) and fourth (both $2.6 \times 10^5$) year of infection. In agreement with the changes in plasma vRNA loads, the mean cell-associated vDNA copy numbers per $10^6$ PBMC increased from $4.4 \times 10^4$ at six wpi to $9.5 \times 10^4$ at 124 wpi in the GU1N group (Supplementary Fig. 2c).

Flow cytometric analyses revealed declines in the absolute numbers of CD4+ T cells during the first year after infection in all four groups (Fig. 2b). At the time of inoculation, all AGMs were juvenile (3–3.5 years old), and it has been reported that CD4+ T cell counts in AGMs decrease with advancing age, even in absence of SIV infection[11,35]. In SIVagm WT infected animals, CD4+ T cell counts declined from baseline values of $1126 \pm 414$ ($n = 12$) to $678 \pm 220$ (first year; $n = 38$, ± gives standard deviation), $412 \pm 209$ (second; $n = 21$), $378 \pm 252$ (third; $n = 21$), and $343 \pm 119$ (fourth; $n = 27$) cells/µl during the 4.7 years of follow-up. The CD4+ T cell counts were even lower in GU1N

infected animals ($541 \pm 33$, $p = 0.006$, $n = 38$; $305 \pm 27$, $p = 0.051$, $N = 21$; $261 \pm 31$, $p = 0.070$, $n = 21$, and $276 \pm 18$ cells/µl, $p = 0.027$, $n = 27$; $p$ values indicate difference to WT infected AGMs and were calculated using Student's $t$-test). However, the baseline values were also higher in WT compared to GU1N infected AGMs ($1126 \pm 414$ vs. $856 \pm 328$ cells/µl, $p = 0.099$, $n = 12$, $t$-test) and the slopes of CD4+ T cell decline did not vary significantly between these groups. Viremic animals from the WT, 1N, and GU1N groups showed significantly lower numbers and percentages of CD4+ T cells than non-viremic AGMs exposed to GU during the final year of follow-up (Fig. 2c). On average, high viral RNA loads correlated with reduced CD4+ T cell counts (Fig. 2d) indicating that efficient replication of both WT and GU1N SIVagm increased the age-dependent loss of CD4+ T cells in AGMs. However, expression of HIV-1-like Vpu and Nef proteins had little, if any, additional effect on CD4+ T cell decline.

**Immune activation and potential correlates of virus control.** To investigate why the four SIVagm constructs replicated with different efficiencies in vivo, we examined various immune indicators of virus infection. All AGMs developed SIVagm specific antibodies against viral Gag and Env proteins, except for the non-viremic animals of the GU group that showed only a weak response against the p27 capsid antigen (Supplementary Fig. 3a). Despite highly varying vRNA levels, the four AGM groups showed similar 4- to 5-fold increases in the levels of urinary neopterin, a marker for immune activation[36], during acute infection (Fig. 3a). A rapid and short-lived 3- to 5-fold increase in CD4+ T lymphocytes expressing CD69 and HLA-DR was observed in AGMs infected with WT SIVagm (Supplementary Fig. 3b). This increase was weaker in GU and 1N infected animals and virtually absent in GU1N infection. In contrast, SIVagm GU1N infection induced a 3- to 5-fold increase in circulating CD8+ CD69+ T cells, while WT SIVagm had only minor effects. Modest increases in circulating Ki67+ CD4+ and Ki67+ CD8+ T lymphocytes were observed between 1 and 20 wpi (Supplementary Fig. 3b), and the single 1N infected AGM 14631 that had undetectable vRNA levels eight wpi showed a strong increase in cytotoxic CD16+ NK cells (Fig. 3b).

All viremic animals showed rapid and strong decreases in circulating CCR5+ CXCR3+ CD4+ T lymphocytes (Fig. 3c). In contrast, non-viremic GU infected AGMs maintained high levels of CCR5+ CD4+ T cells. There was a transient increase in the percentage of CCR5+ CD8+ T lymphocytes in WT infected AGMs that was absent in GU1N infection (Fig. 3c). The levels of CCR5+ CD8+ T cells remained elevated in the two 1N infected AGMs that maintained detectable viral loads after acute infection.

**Fig. 1** SIVagm constructs expressing functional Vpu and HIV-1 Nef. **a** Genomic organization of the WT SIVagm construct with the *nef* gene (yellow) and the GU1N derivative containing the SIVgsn *vpu* (blue) and the HIV-1 NA7 *nef* (orange). **b** HEK293T cells were cotransfected with a *vpu*-deleted HIV-1 NL4-3 construct, expression plasmids for HIV-1 or SIVgsn Vpu and a vector expressing AGM tetherin (BST-2). Infectious virus yield was measured by infection of TZM-bl cells. Curves represent the average values ($n = 3$) relative to those obtained in the absence of tetherin (100%). **c** Levels of AGM BST-2 and human CD4, NTB-A, and CD1d in HEK293T cells cotransfected with SIVgsn or HIV-1 Vpu constructs relative to those measured in cells transfected with the eGFP control vector (100%). Symbols represent the value obtained in one of three experiments. Mean ± SEM are indicated in panels **c**, **d**, **e**, and **j**. **d** Effect of Vpu on NF-κB activity. HEK293T cells were cotransfected with a firefly luciferase NF-κB reporter construct, a *Gaussia* luciferase construct for normalization, a vector for constitutively active IKKβ and constructs expressing eGFP alone or together with Vpu. Luciferase activity was determined two dpi ($n = 3 \pm$ SEM). CTRL represents the vector control. **e** Receptor downmodulation in human PBMCs or CD4+ T cells (CD4) infected with HIV-1 constructs expressing the indicated Nefs relative the *nef*-defective virus (100%), ($n = 3 \pm$ SEM). **f** Effect of NA7 and SIVagm Nefs on virus release in the presence of AGM tetherin in transfected HEK293T cells. Curves represent averages ($n = 3$) relative to those obtained in the absence of tetherin (100%). **g**, **h** Infectivity and replication of SIVagm constructs in vitro. TZM-bl and MOLT-4 clone 8 cells (**g**) or AGM PBMCs (**h**) were infected with the indicated SIVagm constructs. Virus infectivity or production was monitored by β-galactosidase assay or p27 antigen ELISA, respectively. Shown is **g** mean of triplicate infection or **h** one representative experiment. **i** Average infectious virus yield ($n = 3$) from HEK293T cells following co-transfection with SIVagm constructs and plasmids expressing AGM tetherin relative to that detected in the absence of tetherin (100%). **j** Effect of SIVagm constructs on receptor expression by PBMCs from three AGM relative to those infected with the *nef*-defective SIVagm clone (100%). *p*-values were calculated using the two-tailed unpaired Student's-*t*-test and significant differences are indicated as: \*$p < 0.05$; \*\*$p < 0.01$; \*\*\*$p < 0.001$

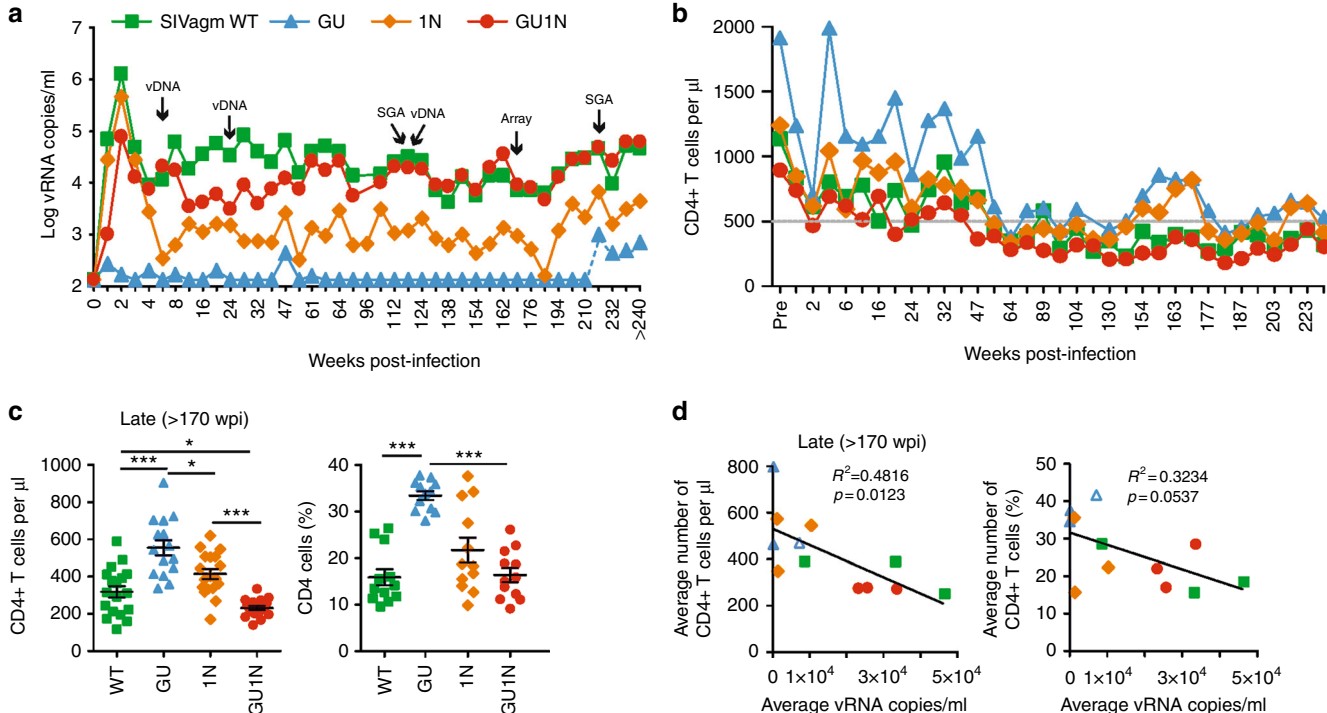

**Fig. 2** Replication of HIV-1-like SIVagm constructs in vivo. **a** Mean levels of plasma viremia in AGMs infected with the indicated SIVagm constructs. Three animals were infected with each virus and plasma RNA loads were determined as described in the Methods section. Arrows indicate time points used for single genome viral RNA analysis (SGA), determination of proviral copy numbers (vDNA), or microarray analysis (Array). Symbols refer to all panels. The limit of viral RNA detection is ~100 copies/ml of plasma. **b** Absolute mean CD4+ T cell counts in AGMs infected with the WT, GU, 1N, and GU1N SIVagm constructs. The prevalue (Pre) represents the average CD4+ T cell count of four measurements prior to infection. **c** Absolute numbers and percentages of CD4+ T cells at later time points of infection (>170 wpi) with mean ± SEM. Each symbol represents one measurement. **d** Correlation between the average vRNA loads and CD4+ T cell counts (left) and percentages (right) in the final year of follow-up. Each symbol corresponds to one AGM and represents the average value of nine measurements from 171 to 240 wpi. The open triangle represents AGM14627 that was accidentally infected. *p*-values were calculated using the two-tailed unpaired Student's-*t*-test and significant differences are indicated as: *$p < 0.05$; **$p < 0.01$; ***$p < 0.001$

AGMs infected with 1N and GU1N constructs expressing the HIV-1 Nef showed higher levels of HLA-DR+ CD4+ and CD8+ T cells in the duodenum at six wpi (Fig. 3d) and in bronchoalveolar lavage (BAL) obtained at weeks 55, 64, and 72 (Fig. 3e) than animals infected with WT SIVagm, suggesting higher levels of immune activation.

Examination of different T cell subsets at late time points (≥170 wpi) showed that WT SIVagm infected animals had higher levels of circulating naive HLA-DR-CD45RA+ CD4+ T lymphocytes and lower levels of memory HLA-DR-CD45RA-CD4+ T cells (Supplementary Figs. 3c, d) compared to GU1N infected AGMs. In contrast, the levels of naive CD8+ T cells were lowest in the WT group (Supplementary Fig. 3c) and AGMs infected with GU1N showed higher levels of activated naive HLA-DR+ CD45RA+ CD8+ T cells (Supplementary Fig. 3e). Increased numbers of naive HLA-DR-CD45RA+ CD4+ T cells in the WT group (Supplementary Fig. 3c, left) suggest that Nef-mediated downmodulation of CD3 was associated with increased levels of CD4+ T cells that did not encounter their respective antigen.

In summary, the four SIVagm chimeras induced different patterns of immune cell proliferation and activation. Strong NK (AGM14631; Fig. 3b) or cytotoxic CD8+ T cell (AGM14626; AGM14630; Fig. 3c) responses may have contributed to the control of viremia in SIVagm 1N infected animals. Despite undetectable vRNA loads, the GU SIVagm constructs induced early immune activation (Fig. 3a), but virus replication remained suppressed throughout the course of the study. Higher levels of activated HLA-DR+ T cells in the duodenum, BAL and blood of the GU1N group (Fig. 3d, e, Supplementary Fig. 3e) suggested

higher levels of immune activation compared to the WT group of AGMs.

**Maintenance of intact *vpu* and *nef* genes**. To study the evolution of the SIVgsn derived *vpu* and HIV-1 derived *nef* genes, we used limiting dilution PCR to amplify 3′ genomic regions (indicated in Fig. 1a) from plasma samples obtained at 121 and 223 wpi. Single genome amplification (SGA) was employed to prevent PCR-induced artifacts, such as recombination, and to assess linkages between *vpu* and *nef* mutations. PCR products were obtained from the plasma of all WT, 1N, and GU1N infected AGMs that had detectable viral loads. However, no viral sequences were amplified from the plasma of AGM14631 as well as all three GU infected monkeys at 121 wpi. When AGM14627 exhibited an unexpected increase in viral load at ≥223 wpi (Supplementary Fig. 2a), we inspected the 223 wpi sequences and found that they encoded the HIV-1 NA7 *nef* allele instead of the SIVgsn *vpu* gene. Moreover, the recovered sequences were closely related to those obtained from AGM14626 infected with the 1N construct (Supplementary Fig. 4b). Both animals were cage neighbors and a documented injury on the hand of AGM14626 suggested that AGM14627 became infected through contact with AGM14626. Amplicons from all other animals contained the expected *vpu* and *nef* alleles and formed animal specific clusters in phylogenetic trees (Supplementary Fig. 4).

Although the various sequences exhibited considerable diversity, the vast majority of amplicons encoded intact *vpu*, *rev*, *env* and *nef* genes. In addition, the SIVgsn *vpu* Kozak sequence

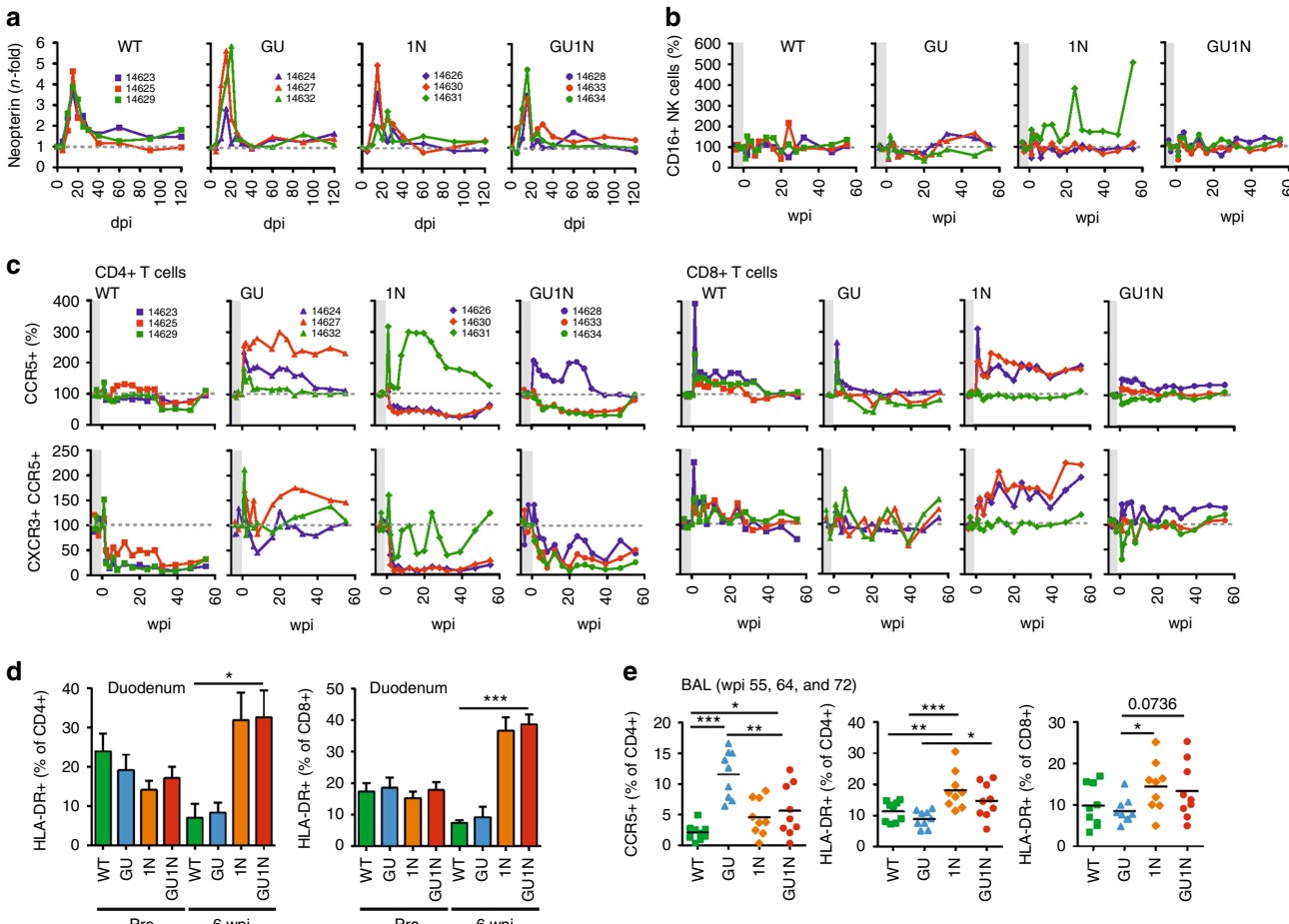

**Fig. 3** Immunological findings during the first year of HIV-1-like SIVagm infection. **a** Increase of urinary neopterin levels at the indicated days post-infection (dpi) relative to the values measured prior to SIV infection. **b**, **c** Levels of CD16+ NK (**b**) and CCR5+ or CXCR3+/CCR5+ CD4+ (left) and CD8+ (right) T cells (**c**) at the indicated weeks post-infection (wpi) relative to the baseline values (100%). The shaded area indicates time points prior to infection. **d** Percentages of activated HLA-DR+ CD4+ and CD8+ T cells in duodenum prior to infection and at six wpi. Shown are mean values obtained for the three animals per group (+SEM). **e** Percentages of CCR5+ CD4+ T cells and activated HLA-DR+ CD4+ and CD8+ T cells in bronchoalveolar lavage obtained at 55, 64, and 72 wpi. $p$-values were calculated using the two-tailed unpaired Student's-$t$-test and significant differences are indicated as: *$p < 0.05$; **$p < 0.01$; ***$p < 0.001$

(GTACGTATG) and the ATTAAGATG sequence upstream of the HIV-1 *nef* gene remained unaltered in most AGMs. Sequences from AGM14631 at 223 wpi were an exception, showing changes to **GC**TAAGATG upstream of the *nef* gene. This 1N infected animal had undetectable plasma viral loads (except for occasional blips) from 12 to 194 wpi, but experienced an increase to ~2000 copies/ml at ≥202 wpi (Supplementary Fig. 2a), raising the possibility that the GC mutation increased viral fitness. Finally, 32 of 33 *nef* alleles obtained from the GU1N infected AGM14634 contained mutations of ACG to ATG restoring the original SIVagm Nef initiation codon and predicting a fusion between the first 56 amino acids of SIVagm Nef and the full-length HIV-1 Nef protein (see 'Alterations in Nef function'). The fact that both HIV-1 *nef* and SIVgsn *vpu* genes remained intact for >4 years strongly suggests that they were advantageous for SIVagm replication in AGMs.

**Long-term maintenance of HIV-1-like Vpu function.** Vpus derived from GU1N infected AGMs at 2 and 4 years post-infection differed only in a few residues from the original SIVgsn71 Vpu (Fig. 4a). Alterations of D31E and G36K/R in the cytoplasmic part emerged in all three animals. Furthermore, F24

changed to Y in AGMs 14628 and 14633 and K39 changed to E or Q in animals 14628 and 14634 and was deleted in 14633. Notably, 13 of 14 substitutions (92%) detected in *vpu* at 121 wpi were non-synonymous, suggestive of selection. To determine possible effects on Vpu function, we tested representative *vpu* genes from the GU1N infected AGMs as well as two synthetic *vpu* alleles containing four or six changes that emerged in vivo (Fig. 4a). All *vpu* alleles were expressed (Fig. 4b) and efficiently counteracted AGM tetherin with Vpus from AGMs 14633 and 14634 exhibiting slightly higher efficiency in enhancing infectious virus yields than the original SIVgsn71 Vpu (Fig. 4c). Furthermore, all AGM-derived Vpus downmodulated human CD4, NTB-A, CD1d, and AGM tetherin and suppressed NF-κB activity (Fig. 4d, e). Thus, Vpu function was retained in AGMs that received the HIV-1-like GU1N SIVagm construct.

**Alterations in Nef function.** An alignment of Nef amino acid sequences from all viremic AGMs (including the infection between cage neighbors) showed that the great majority encoded full-length proteins with previously identified functional motifs remaining intact (Fig. 5a, Supplementary Fig. 5a). In one GU1N infected animal (AGM14634), the original SIVagm *nef* ATG was

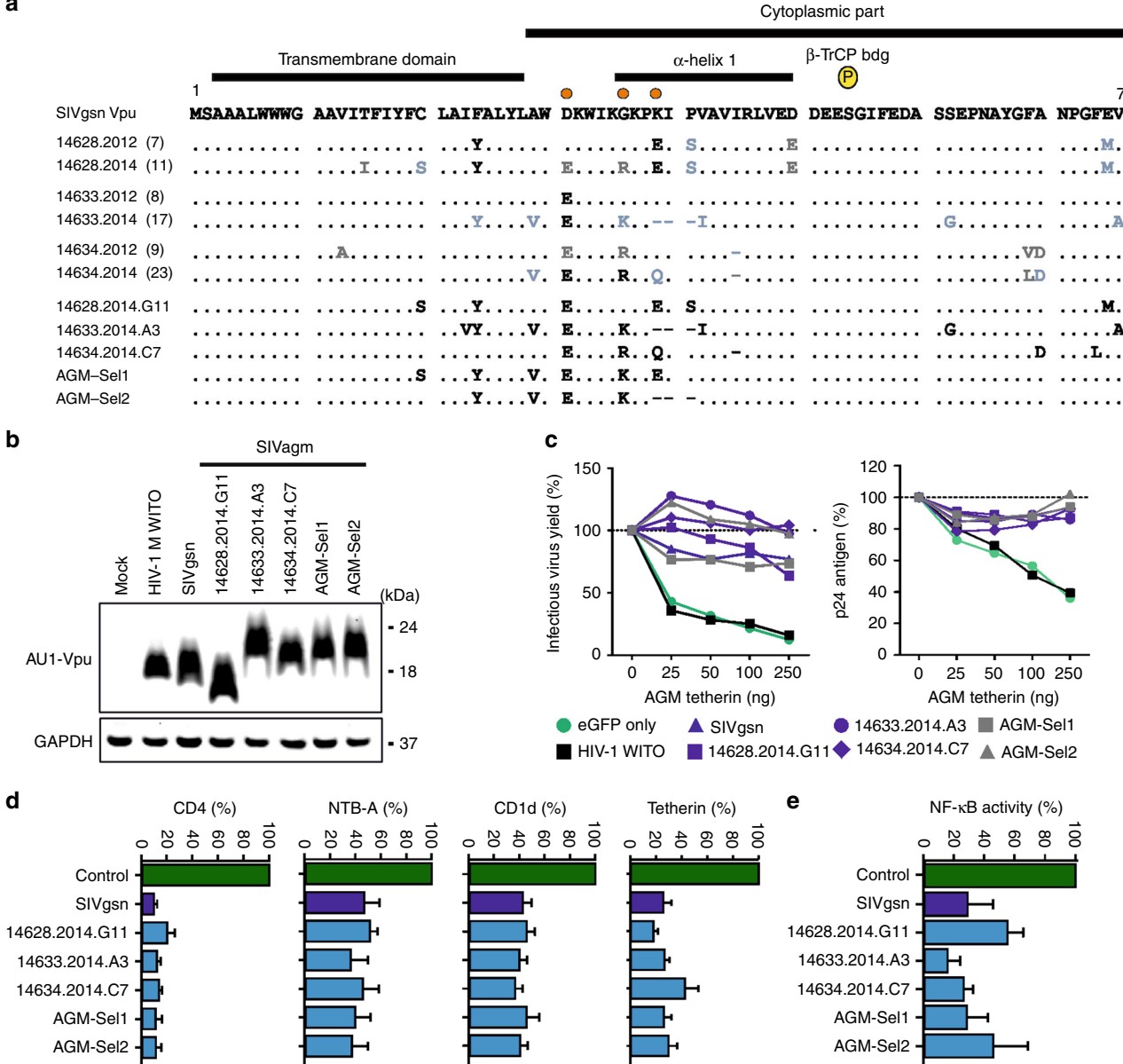

**Fig. 4** Maintenance of HIV-1-like Vpu function in GU1N infected AGMs. **a** Alignment of Vpu sequences derived from GU1N-infected AGMs at 121 (2012) and 223 wpi (2014) and primary or synthetic (Sel1 and Sel2) Vpus chosen for functional analysis. Orange dots highlight mutations found in all three AGMs. Dashes indicate gaps introduced to optimize the alignment. Numbers specify the animal, year of plasma isolation, and number of the respective Vpu sequences. Black indicates changes in all, blue in ≥50% and gray in at least two sequences. **b** Expression of AU1-tagged SIVgsn Vpu proteins. HEK293T cells were transfected with expression plasmids encoding the indicated AU1-tagged Vpus and eGFP. Mock transfected cells were used as negative controls. GAPDH expression levels were analyzed to control for loading. **c** Infectious virus (left) and p24 antigen (right) yield from HEK293T cells cotransfected with an HIV-1 NL4-3 Δvpu construct and vectors expressing the indicated *vpu* alleles in combination with increasing amounts of plasmids expressing AGM tetherin. Shown are average values derived from three experiments relative to those obtained in the absence of tetherin (100%). **d** Vpu-dependent reduction of CD4, NTB-A, CD1d, and AGM tetherin surface expression in HEK293T cells relative to those measured in cells transfected with the eGFP only control vector. Values in panels **d** and **e** represent mean (+SEM) derived from three independent experiments. **e** AGM-derived Vpu proteins suppress NF-κB activity. The effects were measured as described in the legend to Fig. 1c

restored predicting a chimeric protein with 56 SIVagm Nef-derived amino acids fused to the full-length NA7 Nef (Supplementary Fig. 5b). Representative SIVagm and HIV-1 Nefs obtained at 223 wpi were expressed (Fig. 5b) and largely maintained the functional properties of their parental Nef proteins. Importantly, none of the HIV-1 Nefs acquired the ability to downmodulate CD3 (Fig. 5c) or to counteract AGM tetherin (Fig. 5d). All Nef proteins modulated human CD4 (Fig. 5c) and antagonized AGM Serine incorporator 5 (SERINC5) (Fig. 5e).

Several Nef alleles obtained from WT SIVagm infected animals contained changes of G93R, V98A/T and T153/P/E/V (Supplementary Fig. 5a) and displayed increased activity in downmodulating MHC-I compared to the parental SIVagm Nef protein (Fig. 5c). HIV-1 Nef sequences obtained from several 1N or GU1N infected animals contained basic K residues in the acidic domain (EEEE) and/or changes of R8C/S/G and M20K/L. In two of three GU1N infected AGMs, the HIV-1 Nef became less potent in downmodulating CD28, whereas this was not the case

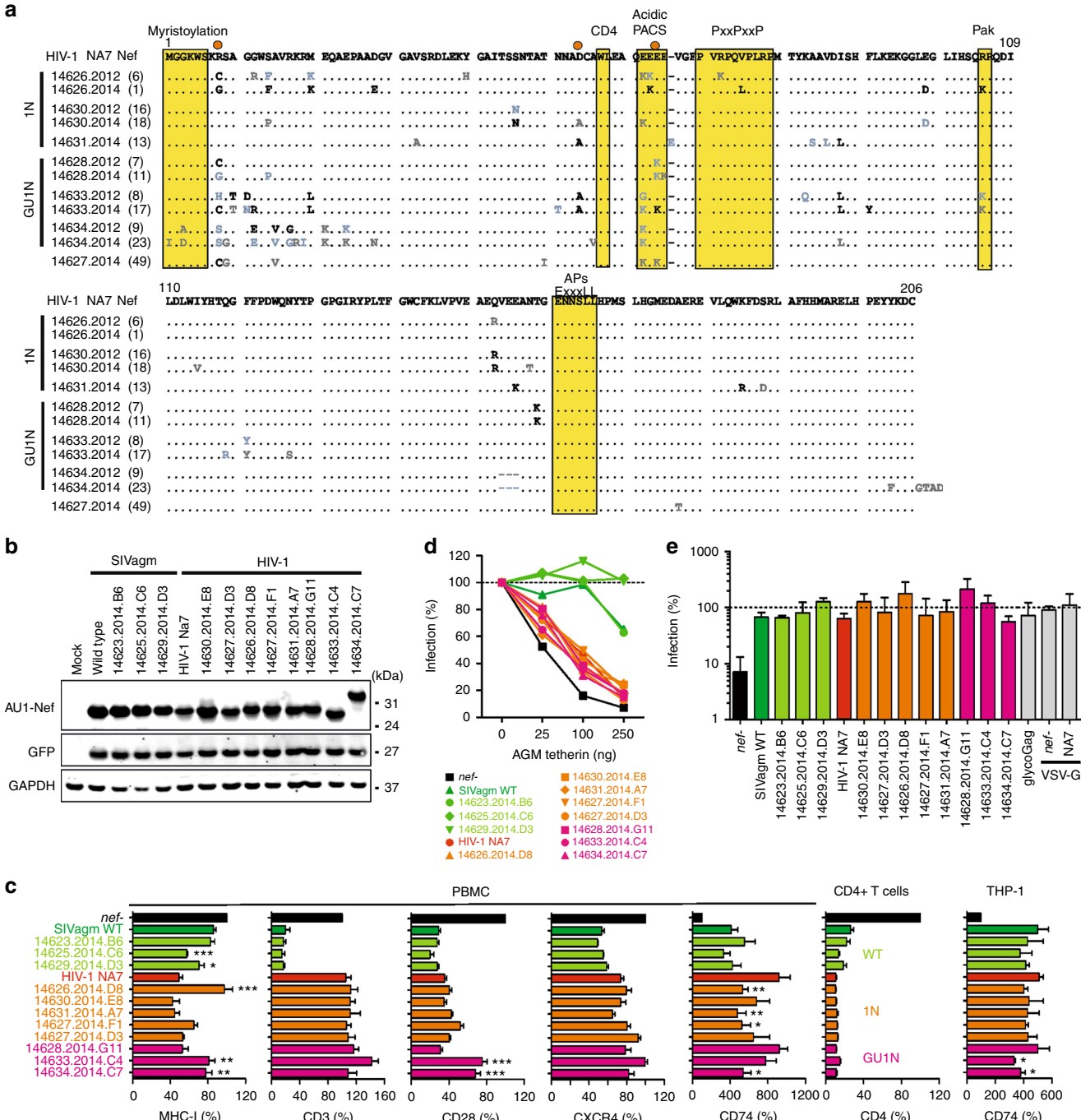

**Fig. 5** Maintenance of HIV-1 Nef function in 1N and GU1N infected AGMs. **a** Alignment of Nef sequences derived from 1N (upper three) and GU1N (lower three) infected AGMs. The HIV-1 NA7 sequence is shown on top and dashes indicate gaps introduced to optimize the alignment. Numbers in parentheses indicate the total number of *nef* alleles analyzed per time point. Black indicates changes in all, blue in ≥50% and gray in at least two sequences. Sites mutated in at least three AGMs are highlighted by orange dots. **b** Expression of selected SIVagm and HIV-1 Nef proteins. HEK293T cells were transfected with expression plasmids encoding the indicated AU1-tagged Nefs and eGFP. GAPDH and eGFP expression levels were analyzed to control for loading and transfection efficiency, respectively. **c** Quantitative assessment of Nef-mediated downmodulation of the indicated cellular receptors on PBMCs (MHC-I, CD3, CD28, CXCR4, and CD74), CD4+ T cells (CD4) and THP-1 cells (CD74). Shown are mean (+SD) fluorescence intensities (MFIs) of receptor expression relative to the *nef*-defective control (100%) derived from three experiments. **d** HIV-1 Nefs did not evolve activity against AGM tetherin. HEK293T cells were cotransfected with HIV-1 NL4-3 *nef-vpu-* constructs, expression plasmids for the indicated *nef* alleles and different amounts of AGM tetherin. Infectious virus yield in the culture supernatants two dpi was determined by triplicate infection of TZM-bl reporter cells. Shown is the mean of three experiments. **e** Antagonism of AGM SERINC5 by SIVagm and HIV-1 Nef proteins. HEK293T cells were cotransfected with HIV-1 NL4-3 proviral constructs containing the indicated *nef* alleles and an empty vector control or AGM SERINC5 expression vector. Viral supernatants were obtained 3 days later and infectious HIV-1 yield determined by triplicate infection of TZM-bl cells. Shown are average values of three experiments + SD (*n* = 3) relative to those obtained in the absence of SERINC5 expression vector (100%)

in AGMs infected with the 1N construct (Fig. 5c). The SIVagm-NA7 Nef fusion that emerged in the third animal (AGM14634) was active in all in vitro assays, albeit less so than the original Nef in modulating MHC-I and CD28. Thus, except for the loss of CD28 downmodulation activity in GU1N infected AGMs, HIV-1-like Nef functions were well preserved in all animals in vivo.

**HIV-like SIVagm infection induces immune activation pathways.** To compare the effects of WT and chimeric SIVagm on cellular gene expression and systemic immune activation in vivo, we performed a cross-sectional analysis of gene expression using Affymetrix Rhesus Genome microarrays and RNA from whole blood of the 12 AGMs at 170 wpi. We selected this array because rhesus macaques represent the species most closely related to AGMs with a commercial GeneChip available. Previous studies have shown that it is suitable for analyses of AGMs[37] and more distantly related sooty mangabeys[38]. Calculation of GeneChip hybridization intensities confirmed that those obtained for AGMs were similar to those measured using macaque samples. Due to the small sample size, we employed gene set enrichment analysis (GSEA)[39], rather than conventional statistics, to identify signaling pathways significantly altered by the chimeric viruses relative to WT infection. In previous studies, we derived several sets of genes that discriminated natural SIV infection of sooty mangabeys from non-natural infection of rhesus macaques during the chronic infection phase[38]. One of these blood-derived gene signatures (SIV chronic immune activation) was not only associated with infection of rhesus macaques versus sooty mangabeys, but also correlated with rapid disease progression (rather than viremic non-progression) in HIV-1 infected humans[40]. Interestingly, the SIV chronic immune activation gene set was significantly enriched in animals infected with the GU1N virus compared to WT SIVagm infected monkeys (Fig. 6a, b). Similarly, a panel of inflammation-related genes (inflammation and immune activation) defined previously as being modulated after IL-21 treatment of SIV-infected macaques[41] was significantly enriched in the GU1N group compared to WT infected animals (Fig. 6a, c). To test more broadly for signals of immune activation, we performed GSEA with gene sets from the KEGG immunity collection. We found that some pathways associated with adaptive immune signaling (T Cell Receptor, B Cell receptor, Th1, and Th2) were enriched in both GU and GU1N relative to WT infection (Fig. 6d). Histological analyses (see 'Histological and pathological findings at necropsy') detected virally infected cells in non-viremic SIVagm GU exposed AGMs. Thus, persistent viral gene expression in lymphoid tissues of these animals might induce some immune activation. However, several pathways associated with innate immunity (NLR-, RLR-, TLR-signaling, and cytosolic DNA sensing) were only enriched in the GU1N group (Fig. 6d). Collectively, these data suggest that infection of AGMs with the GU1N virus induces transcriptional pathways associated with immune activation above the levels seen in WT SIVagm infection.

**Histological and pathological findings at necropsy.** All 12 AGMs were euthanized between 244 and 255 weeks post-infection. Sections from the gut (jejunum) collected at necropsy were analyzed for the presence of viral RNA by FISH. Cells producing viral RNA were detected in animals from all four groups (examples shown in Fig. 7). To verify the presence of SIV+ cells in tissues from the non-viremic GU group, we also stained sections from the ileum and duodenum. Our analyses confirmed the presence of viral RNA producing cells in the small intestine of AGMs 14624 and 14632 despite undetectable viral RNA loads during the last 3 years of follow-up (Supplementary Figs. 2A and 6). Examination at necropsy failed to reveal gross lesions or

opportunistic infections. Histologically, several animals suffered from a mild enterocolitis that was not associated with diarrhea and observed in all four AGM groups, irrespective of the levels of viral replication. Thus, none of the SIVagm constructs, including the GU1N chimera, caused immunodeficiency within 4.7 years of infection.

To examine possible inflammatory events in the intestinal tract, we stained jejunum fragments from one randomly selected animal from each group for IL-18, a potent pro-inflammatory cytokine. Representative images of these immunohistochemical analyses are shown in Fig. 8. We detected IL-18+ cells in the top of the villi in all four AGMs. In the crypt, however, IL-18 positive cells were only observed in the GU1N-infected AGM14634 (Fig. 8). IL-18 production in the abortively infected AGM14624 was low. While more comprehensive quantitative analyses in a larger number of animals are needed to draw definitive conclusions, these results suggest modest differences in small intestine inflammation levels between GU1N and WT SIVagm infected AGMs.

## Discussion

Over many hundreds of thousand years of coevolution some simian "immunodeficiency" viruses have achieved a well-balanced benign relationship with their hosts: they replicate to high titers, infect a large proportion of animals (often >50%), but fail to cause CD4 T cell depletion and disease. In contrast, SIVs that ultimately contributed to the emergence of HIV-1, such as SIVs infecting greater spot-nosed, mona and mustached monkeys, are much less common and widespread (usually <2%)[42-44] and the ape precursor of HIV-1 causes AIDS in its natural chimpanzee host[23]. This altered virus-host relationship seems to be due to a series of comparably recent evolutionary events that appear to be associated with increased viral pathogenicity, beginning with the acquisition of a *vpu* gene by the precursor of SIVs nowadays infecting various *Cercopithecus* monkey species[6]. A subsequent key event that actually occurred twice during the cross-species transmission and recombination events that gave rise to other *vpu* containing lentiviruses, such as SIVcpz and ultimately HIV-1, was the complete loss of the TCR-CD3 downmodulation function of Nef[6,45,46]. Here, we aimed to recapitulate these events by first inserting a *vpu* gene into a non-pathogenic SIV backbone and next combining it with a Nef lacking the CD3 modulation function. Our goals were to determine (i) why the acquisition of *vpu* triggered changes in Nef function and (ii) whether these intrinsic properties of HIV-1 increase viral pathogenicity. We show that *vpu* can only promote lentivirus replication in combination with a Nef that is inactive against CD3. This may explain why this Nef function was lost twice within the HIV-1/SIVcpz lineage[6]. In addition, we show that HIV-1 intrinsic Vpu and Nef functions are indeed associated with increased systemic immune activation and thus appear to contribute to the virulence of HIV-1 in humans. However, these viral properties did not cause accelerated CD4+ T cell depletion and immunodeficiency in AGMs, indicating that these intrinsic viral properties are not sufficient to counteract effective host protective mechanisms.

It has been shown that an unusual variant of SIVsmm (PBj14) causes rapid morbidity and mortality in both macaques and sooty mangabeys, the latter of which are usually resistant to SIV-induced disease[47]. The main pathogenicity determinant of this virus was identified to be a Nef protein that causes extensive T cell activation[22]. HIV-1 Nef proteins do not activate T lymphocytes on their own. However, in contrast to Nef proteins that down-modulate the TCR-CD3 complex, they boost instead of diminish the responsiveness of infected T cells to exogenous

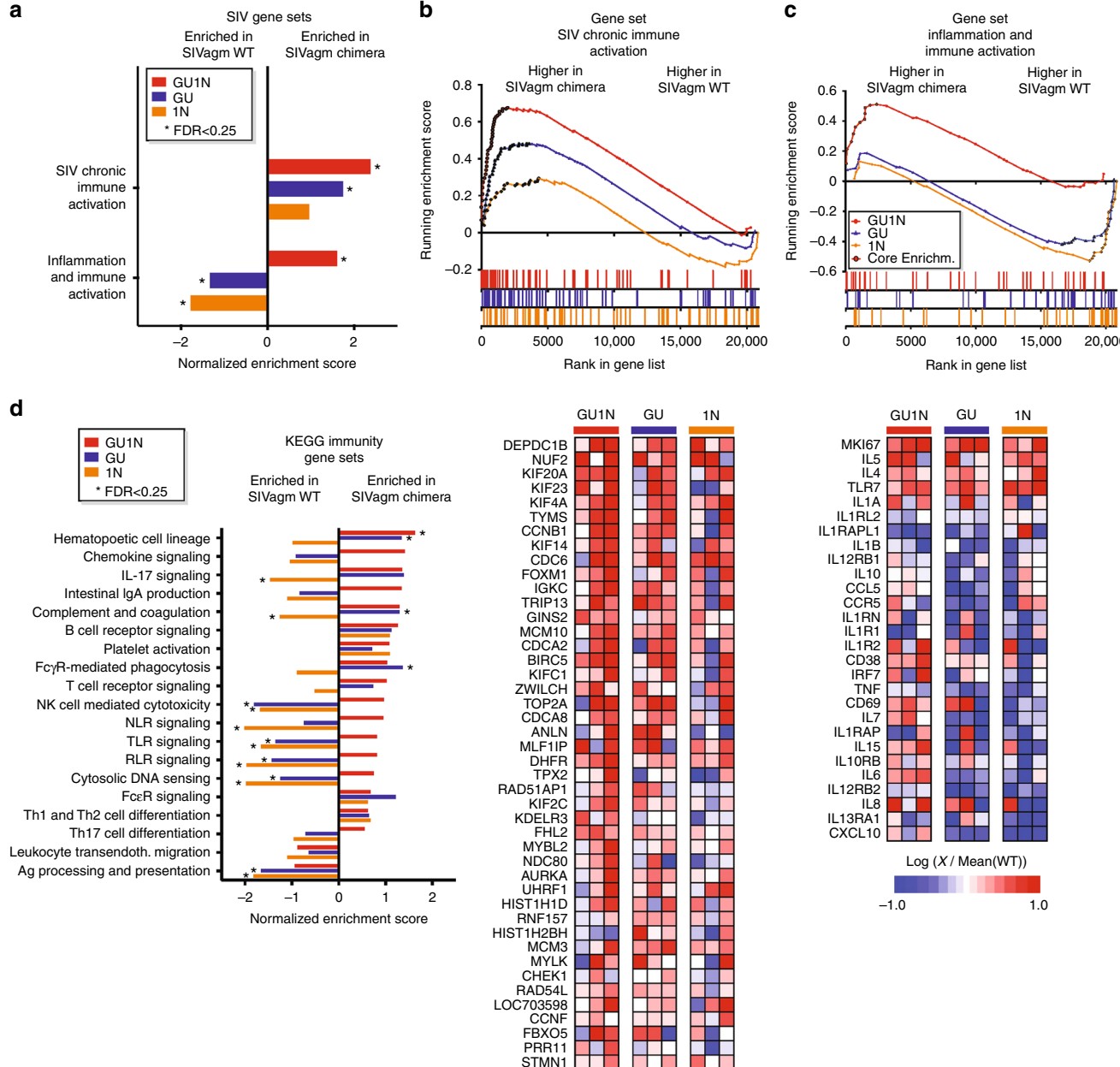

**Fig. 6** Effect of WT and chimeric SIVagm infection on cellular gene expression in AGMs. **a, d** Enrichment scores of gene sets generated from in vivo SIV infection transcriptional profiling experiments (**a**) or immune-related genes from the KEGG database **d**. GSEA analysis was used to contrast transcriptome-wide gene expression profiles from day 170 between the WT SIVagm infected samples and each of the chimeric infections. Gene sets with an FDR of <0.25 were considered significant and are indicated by an asterisk. **b, c** Enrichment plot (upper) and leading-edge heat map (lower) of GSEA analysis of WT vs. GU1N, WT vs. GU, and WT vs. 1N datasets for enrichment of the "SIV chronic immune activation" (**b**) and "inflammation and immune activation" (**c**) gene sets. The plot in the upper panels depicts the running enrichment score indicated by the line adjoining data points. Each data point represents an individual gene in the gene set. Genes contributing the most to the enrichment score (Core genes or leading edge genes) are indicated by bolded symbols. Each bar in the stick plot at bottom indicates the position of an individual gene from the gene set within the entire dataset (indicated on X axis). The lower panel depicts a heat map of "leading edge" or "Core" genes from the enrichment plots. The genes in the leading edge of all three GSEA contrasts (WT vs. GU1N, WT vs. GU, and WT vs. 1N) were pooled, and normalized to the mean of the WT genes. Genes are ranked according to their position in the WT vs. GU1N analysis. Color scale shown at bottom

stimulation[18,48,49]. Thus, the HIV-1 Nef represents a phenotypic intermediate between the Nefs of the highly pathogenic SIVsmmPBj14 strain and nonpathogenic SIVs. In support of this, HIV-1 is more pathogenic than HIV-2 in humans[50] and SIVcpz causes disease in the wild[23]. It has been shown that Nef-mediated downmodulation of TCR-CD3 suppresses the expression of inflammatory cytokines by virally infected primary CD4+ T cells[21,51]. In agreement with these in vitro data, different gene

sets associated with chronic immune activation and inflammation were expressed at higher levels in AGMs infected with the HIV-1-like chimeric virus, although the effects were modest compared to what has been observed in HIV-1 infected humans. The reason for this is that well-adapted hosts of SIV, such as African green monkeys and sooty mangabeys, have developed mechanisms to protect critical CD4+ T cell subsets from virus infection[1,11,35,52]. Limited infection of central memory CD4+ T cells has also been

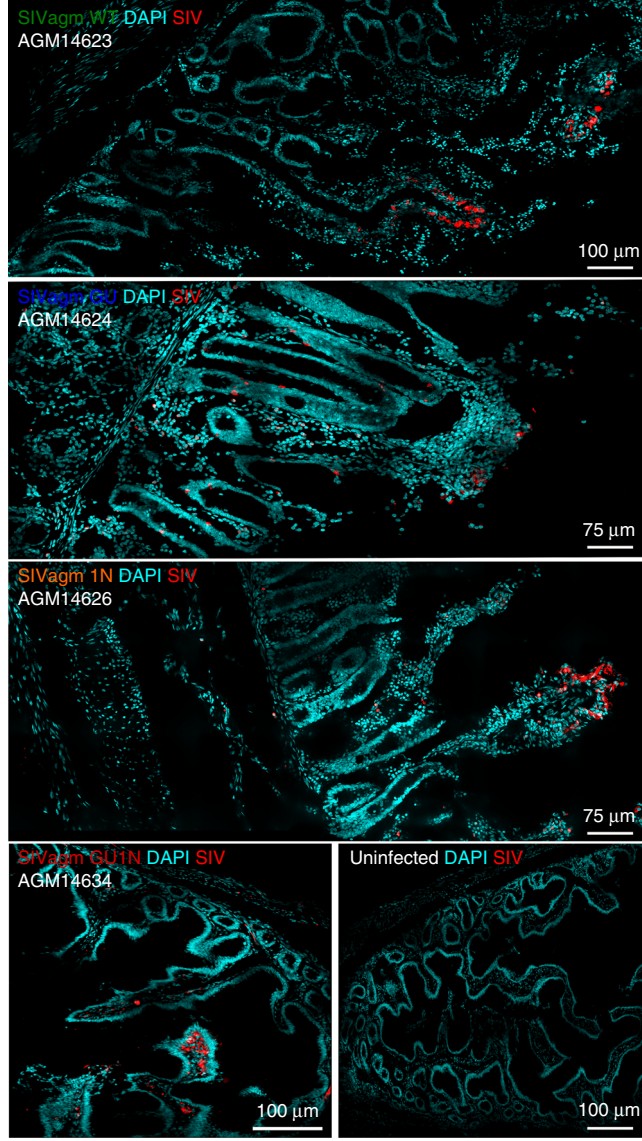

**Fig. 7** SIV distribution in AGM jejunum sections during the chronic phase of infection. Confocal images of SIV-infected cells (red), nucleus (blue) in Jejunum for each group are shown. Representative SIV-specific in situ hybridization during chronic infection, demonstrating productively infected cells in jejunum for each virus. Jejunum sections from an uninfected monkey were used as control to demonstrate the specificity of the SIV probes. Images derived from mounted multiple confocal projected z-scans. Pictures were obtained using a Leica SP8 confocal microscope and processed with ImageJ software

demonstrated in a very rare subset of HIV-1 infected individuals who maintain normal levels of CD4+ T-cells despite persistently high viremia[53]. Furthermore, recent analyses identified variations in several immune-related genes in sooty mangabeys that may contribute to the lack of harmful chronic inflammation in this natural host of SIV[12]. Thus, it seems clear that some viral pathogenicity determinants promote disease progression, but only in incompletely adapted (more recently infected) hosts, such as chimpanzees and humans[21,54]. In this context, it will be important to determine whether the attenuated replication of the SIVagm GU1N variant during the first year of infection slowed disease progression and whether in vivo passage resulted in adaptive changes that increase its replication and disease causing potential.

One key question of our study was to what extent the various SIVagm constructs are able to initiate and maintain a productive infection in African green monkeys. Although we expected some Vpu and Nef interaction, the finding that *vpu* alone rendered SIVagm entirely non-infectious in vivo, but not in cell culture, came as surprise. The fact that only their combination promoted efficient replication of SIVagm indicates that HIV-1-like Vpu and Nef proteins are functionally linked. In fact, HIV-1 Nef boosts the activation of NF-κB early during the viral life cycle to initiate efficient proviral transcription, while Vpu suppresses NF-κB-dependent antiviral gene expression at later stages[15]. In contrast, SIVagm and most other lentiviruses lacking *vpu* use Nef-mediated CD3 downmodulation as an alternative strategy to suppress T lymphocyte activation and to minimize antiviral gene expression during later stages of their replication cycle[55]. It is tempting to speculate that the SIVagm GU construct failed to replicate efficiently in vivo because the combination of both suppression of NF-κB activity and CD3 downmodulation may reduce the state of T cell activation below the threshold required for effective viral gene expression. However, occasional blips in viral RNA (Supplementary Fig. 2A), modest humoral immune responses against the p27 core antigen (Supplementary Figure 3A), and detection of SIV+ cells in lymphoid tissues (Fig. 7 and Supplementary Fig. 6) suggests that the SIVagm GU construct persisted at low levels in lymphoid organs.

Another unexpected result was that the SIVagm 1N construct showed high levels of viremia during acute infection despite the lack of an effective tetherin antagonist, while viral loads during chronic infection were about 20-fold lower than in animals infected with wild-type SIVagm or the HIV-1-like double chimera. This is consistent with the finding that LP-BM5 murine leukemia virus (MuLV) showed similar levels of replication in wild-type and tetherin knock-out mice during acute infection but developed higher viral loads in the absence of tetherin at later time points[56]. We cannot exclude the possibility that Env gained some anti-tetherin activity, as reported for some HIV-2 and SIV strains[57–59]. Nonetheless, the high levels of viremia 2 weeks post-infection suggest that effective tetherin antagonism might not be critical during acute SIV infection of AGMs, although the markedly reduced viral loads in chronic SIVagm 1N infection support a role of tetherin antagonism during later stages. Despite relatively low viral loads of ~2000 copies per ml plasma, AGM14626 apparently transmitted SIVagm 1N to its cage neighbor AGM14627 (the two animals were housed in separate but adjoining cages). Assuming no anti-tetherin activity of any other viral gene product, these results suggest that tetherin antagonism may not be absolutely required for SIV transmission.

In conclusion, we show that HIV-1 intrinsic Vpu and Nef properties can be implanted into SIVagm and that their combination allows efficient viral replication in African green monkeys. Vpu-mediated tetherin counteraction and lack of CD3 downmodulation by Nef were maintained over almost 5 years of follow-up. In combination with an HIV-1 *nef* gene, Vpu increased viral loads during chronic infection by more than one order of magnitude. Thus, we have created the first non-human primate model that faithfully recapitulates characteristic accessory gene functions of HIV-1 infection in humans. This model can be used to dissect which of the multiple Vpu functions, such as CD4 degradation, tetherin antagonism[13,14], suppression of NF-κB activity[15], and/or NK cell evasion[60], are essential in vivo. Finally, NF-κB plays a key role in viral latency[61,62] and HIV-1 uses both Nef and Vpu to fine-tune the activity of this transcription factor throughout its replication cycle[15]. Thus, our study provides an in vivo framework to examine whether these HIV-1 specific accessory gene functions affect the establishment, maintenance, and/or reactivation of the latent virus reservoirs.

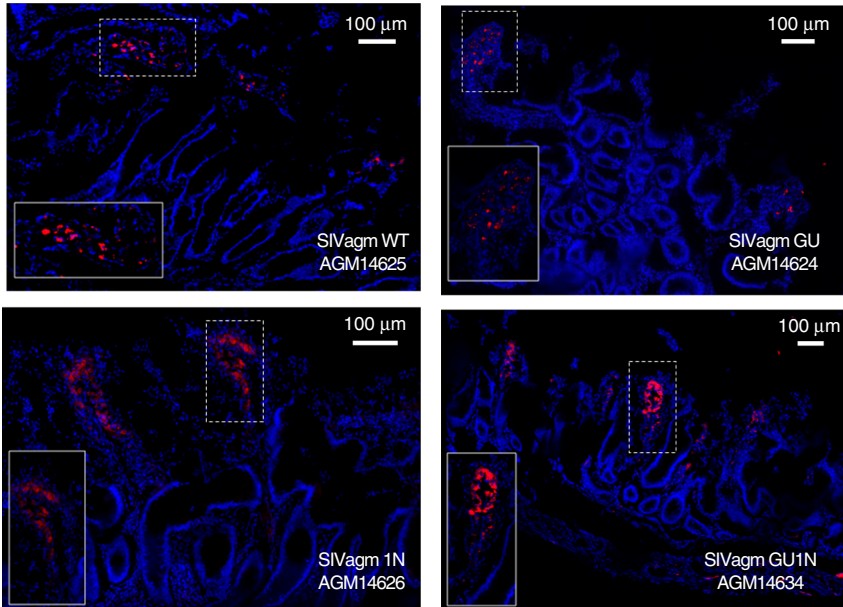

**Fig. 8** Detection of IL-18 production in the gut. Sections of jejunum from the indicated animals were stained for IL-18 (red) and DAPI total nucleus (blue) and analyzed by confocal microscopy. The picture represents the distribution of IL-18+ cells in the gut of one chronically infected AGM per group. A representative picture out of 2–3 sections taken per monkey is shown

## Methods

**Animals**. The African green monkeys (*Chlorocebus aethiops sabeaus*, AGM) used in this study were cared for by experienced staff of the German Primate Center (DPZ) and housed according to the German Animal Welfare Act, which complies with the European Union guidelines on the use of non-human primates for biomedical research and the Weatherall report. The study was approved by the Lower Saxony State Office for Consumer Protection and Food Safety and performed with the project licenses 33.9.42502-04/094/08 and 33.4-42502-04-12/0970. In line with §11 of the German Animal Welfare act, the DPZ has the permission to breed and house non-human primates under license number 392001/7 granted by the local veterinary office. If compatible with each other, monkeys were housed in groups of two by combining cages or, if not, in single cages. Animals that had to be caged individually had constant visual, olfactory and acoustic contact to their roommates and could still groom their neighbors through small mash inserts in the separating sidewalls. Each cage was equipped with a perch. The animals had water access ad libitum and were fed with dry monkey biscuits containing adequate carbohydrate, energy, fat, fiber (10%), mineral, protein, and vitamin content twice daily. The feed was supplemented by fresh fruit or vegetables and varying treats like nuts, cereal pulp, and different seeds to make foraging more attractive. Moreover, for environmental enrichment feeding puzzles, alternate toys and wood sticks for gnawing were offered. During the study, animals were assessed by experienced animal caretakers twice a day for any signs of distress, pain, or sickness by checking water and feed intake, feces consistency, and general condition. In case of any abnormal presentation, animals were attended by veterinarians.

**Cell culture**. HEK293T (ATCC, Cat# CRL-3216) and TZM-bl (NIH, Cat# 8129) cells were cultured in Dulbecco modified eagle medium (DMEM) supplemented with 10% (v/v) FBS, 2 mM L-glutamine, 100 µg/ml streptomycin and 100 U/ml penicillin (all from Gibco). MOLT-4 clone 8 (NIH, Cat# 175) and THP-1 (ATCC, Cat# TIB-202) cells were cultured in supplemented Roswell Park Memorial Institute (RPMI) 1640 medium. All cells grew at 37 °C with 5% $CO_2$ and were tested routinely for mycoplasma contamination by a PCR-based method. The cell lines were authenticated by ATCC or NIH and not validated further in our laboratory.

PBMCs from healthy human donors were obtained from lymphocyte concentrates from 500 ml whole blood (blood bank Ulm) and PBMCs from fresh blood of healthy AGMs (Silabe, Strasbourg) by density centrifugation. For viral infection, cells were pre-stimulated with 1 µg/ml phytohemagglutinin (PHA) and 10 ng/ml human recombinant interleukin-2 (IL-2) for 3 days. Human CD4+ T cells were enriched from buffy coats through negative selection using the RosetteSep™ Human CD4+ T Cell Enrichment Cocktail (Stemcell) according to manufacturer's protocol and stimulated with 10 ng/ml IL-2 and Dynabeads® Human T-Activator CD3/CD28 in a ratio of 1:1.

**Viruses and transfection**. All infections were conducted using SIVagm Sab92018ivTF[24] or derivatives thereof and HIV-1 NL4-3 constructs with or without an IRES element and the *eGFP* gene[18]. Virus stocks were generated by

transient transfection of HEK293T cells using the calcium phosphate precipitation method. For infection of pre-stimulated human PBMCs, CD4+ T cells, or Molt-4 clone 8 cells, the cells were infected with unmodified or VSV-G pseudotyped virus stocks containing 50 ng of p27 antigen or by employing 300 µl of virus stock for one Mio cells. Pre-stimulated AGM PBMCs were infected using the spinfection protocol. Briefly, one million cells were seeded in 96-well plates (U-Bottom), 100 µl of the SIVagm virus stocks was added and spinfection was performed at 1200 rcf for 2 h at 37 °C. Cells were then washed and transferred in 6-well plates with 2 ml supplemented RPMI-1640 with 10 ng/ml IL-2. Three days post-infection, cells were stained for FACS analyses. All infection experiments were performed by trained staff (who received safety instructions according to GenTSV §12, Abs. 3,5 S3+ Infektionsschutz, German law) in an approved biosafety level-3** (BSL-3**) laboratory. Work was approved by the Regierungspräsidium Tübingen, Germany.

**Antibodies and reagents**. For flow cytometry, following antibodies were titrated and validated using matching isotype controls from the same supplier: Anti-HIV-1 core antigen (1:100, KC57, Beckman Coulter, Cat# 6604665), anti-human CD4 (1:40, S3.5, Invitrogen, Cat# MHCD0405), anti-human BST2 (1:20, RS38E, Bio-Legend, Cat# 348410), anti-human NTB-A (1:20, 292811, R&D Systems, Cat# FAB19081A), anti-human CD1d (1:10, CD1d42, BD Biosciences, Cat# 550255), anti-human CD28 (1:10, L293, BD Biosciences, Cat# 348047), anti-human MHCI (1:10, W6/32, BioLegend, Cat# 311410), anti-human/simian CD3 (1:50, SP34-2, BD Biosciences, Cat# 560351, Cat# 557749 or Cat# 557917), anti-human CD74 (1:33, M-B741, Ancell, Cat# 226-050), anti-human CXCR4 (1:5, 12G5, BD Biosciences, Cat# 555976), anti-human CD4 (1:20, RPA-T4, BD Biosciences, Cat# 555349), anti-human/simian MHCI (1:10, G46-2.6, BD Biosciences, Cat# 555553), anti-human/simian CD4 (1:50, L200, BD Biosciences, Cat# 560837 or Cat# 560811), anti-human/simian CD11a (1:25, HI 111, BD Biosciences, Cat# 550852), anti-human/simian CD11b/Mac-1 (1:100, ICRF44, BD Biosciences, Cat# 558123), anti-human/simian CD14 (1:50, M5E2, BD Biosciences, Cat# 550787), anti-human/simian CD16 (1:25, 3G8, BD Biosciences, Cat# 557758), anti-simian CD45 (1:50, MB4-6D6, Miltenyi Biotec, Cat# 130-091-898), anti-human CD183 (1:17, 1C6/CXCR3, BD Biosciences, Cat# 557185), and anti-human CD195 (1:11, 3A9, BD Biosciences, Cat# 550856).

For immunoblot analysis, we employed anti-AU1 (1:10,000, AU1, Covance, Cat# MMS-130R), anti-GFP (1:3,000, polyclonal, Abcam, Cat# ab290), anti-GAPDH (1:200, polyclonal, BioLegend, Cat# 631401) and secondary anti-mouse or anti-rabbit IRDye Odyssey antibodies (1:20,000, LI-COR, Cat# 926-32220, Cat# 926-32211 or Cat# 926-32221) or alkaline phosphatase-conjugated anti-human IgG (1:1,000, Jackson ImmunoResearch, Cat# 109-055-098).

For histological analysis, anti-human IL-18 (1:500, polyclonal, Atlas antibodies, Cat# HPA003980), goat anti-rabbit Alexa Fluor 568 (1:800, Abcam, Cat# ab175695), and DAPI (0,05 µg/ml, Sigma Aldrich, Cat# D9542) were used.

**Flow cytometry**. A volume of 50 µl whole blood, 500,000 PBMCs, or 300,000 mononuclear cells (MNCs) purified from BAL or duodenum[63] were stained for 30 min at room temperature in the dark with mixtures of pre-titrated antibodies. For

intracellular staining, a Fix and Perm kit (Biozol) was used according to manufacturer's instructions. Briefly, the cells were washed and fixed after staining of the surface receptors with Fixing Solution A. The cells were then washed and incubated with the Permeabilization Solution B containing the antibody used for intracellular staining for 30 min at 4 °C. FACS analyses was performed with a LSR II flow cytometer or a FACSCantoII (BD Biosciences) and data were analyzed using the BD FACSDIVA™ software.

**Determination of CD4+ T cell counts**. CD4+ T cell proportions were determined by staining whole-blood leukocytes with a pre-titrated antibody cocktail comprising anti-CD11a-allophycocyanin (APC), CD3-Alexa700 and CD4-Horizon V450. Following staining with the antibody mixture, red blood cells were lysed with BD FACS lysing solution and labeled lymphocytes were analyzed for their expression of cell surface markers by flow cytometry on an LSR II flow cytometer (BD Biosciences). Lymphocyte populations were gated based on forward and side scatter characteristics and then exclusion of doublets and expression of CD11a, followed by that of CD3. Data were generated with BD FACS Diva 6.1.3 Software before analysis with FlowJo 8.8 Software (Treestar).

**Generation of HIV-1 like SIVagm constructs**. To insert the SIVgsn *vpu* into the SIVagmSab92018 molecular clone[24], the AflII and PstI restriction sites at nucleotide positions 4160 and 4520 were eliminated by splice overlap extension (SOE)-PCR. A chemically synthesized 1041 bp fragment containing parts of *tat*, the SIVgsn *vpu* gene, and the 5′ end of *env* flanked by PstI and AflII sites was cloned into the unique AflII and PstI restriction sites of the modified SIVagm Sab92018ivTF construct. To replace *nef*, the original AGM *nef* start codon was mutated, thereby eliminating the overlap between the *env* and *nef* reading frames. A synthetic 1350 bp DNA fragment encompassing the 3′end of *env*, the HIV-1 NA7 *nef* gene and the SIVagm 3′LTR was cloned into SIVagmSab92018ivTF using the unique EcoRI restriction site in *env* and the MluI site flanking the 3′LTR. All chimeric SIVagm constructs were sequenced to verify the presence and integrity of heterologous *vpu* and *nef* alleles.

**AGM infection and specimen collection**. Twelve purposed-bred healthy juvenile female AGMs (*Chlorocebus sabaeus*) were imported from Barbados. When entering the experiment they were 3–3.5 years old with a bodyweight (BW) between 2.7 and 3.4 kg. All monkeys were seronegative for Simian Retrovirus Type D, Simian Immunodeficiency Virus, and except for 14,624, Simian T-lymphotropic Virus Type 1. Animals were randomly distributed to four groups with three monkeys each. All manipulations took place in the morning for which the animals had to fast for 18 h beforehand. To take blood samples from the femoral vein by using the vacutainer system (BD), animals were anesthetized by intramuscular (i.m.) injection of 10 mg ketamine per kilogram body weight. For virus inoculation, medical interventions like collecting BAL and intestinal biopsies, removal of peripheral lymph nodes, and as premedication for euthanasia, all for which a deeper anesthesia with appropriate analgesia was required, animals were administered i.m. a mixture of 5 mg ketamine, 1 mg xylazine and 0.01 mg atropine per kg BW. At necropsy, animals were euthanized by an overdose of 160–240 mg sodium pentobarbital per kg BW injected into the circulation. Groups of three randomly distributed animals each were inoculated intravenously with 1 ml of cell-free culture supernatant containing 500 ng of p27 capsid antigen of the respective SIVagm construct as described for the SIVagmSab92018ivTF clone[24]. Blood samples were collected twice or thrice before infection, in 2–4-week intervals until 32 weeks post infection (wpi) and thereafter around every 8 weeks until the end of the study (weeks 244–255). Duodenal biopsies were sampled by endoscopy three times before infection and seven times until 70 wpi and BAL was performed between 55 and 72 wpi as reported[63]. Peripheral lymph nodes were surgically removed once before infection and four times between 2 and 64 wpi.

**Single-genome amplification**. Viral RNA was extracted from the plasma of infected AGMs using the QIAamp viral RNA minikit (Qiagen). cDNA was synthesized using SuperScript III reverse transcriptase (Invitrogen) and the SIVagm strain-specific primer hb3′r1 (5′-GCGAACACCCAGGCTCAAGCTG-3′). Single genome amplification was performed as previously described[24], using hb3′r1 (5′-GCGAACACCCAGGCTCAAGCTG-3′) and sab3′f1 (5′-CAAATGGATTGTA-CACACCTGG-AAGGAAA-3′) in the first round of PCR, and newsab3′r1 (5′-ACGGGGTAAGCCACTCCCAGTAC-3′) and sab3′f2 (5′-TGTTGGTGGGGAAAGATAGAGC-ACTC-3′) in the second round of PCR. PCR amplicons were directly sequenced using an ABI 3730 DNA analyzer for 121 wpi samples and MiSeq Illumina for 223 wpi samples.

**Cloning of mutant *vpu* and *nef* alleles**. Expression vectors carrying different *vpu* and *nef* alleles were generated by cloning into a pCG expression vector co-expressing eGFP[27]. The required XbaI and MluI flanking the reading frames and an AU1 tag (GACACCTATCGCTATATATA) were introduced to the SGA products by SOE-PCR. To generate HIV-1 (NL4-3 based) constructs encoding various *nef* genes, SOE-PCR was performed to introduce HpaI and MluI restrictions sites into the AGM *nef* fragments generated by SGA. These fragments were then cloned into pBR_NL4-3 IRES eGFP[18] and pBR_NL4-3 and pBR_NL4-3 (*vpu* STOP) (*env*

STOP) IRES eGFP[64], respectively, using the unique HpaI and MluI sites. To generate HIV-1 NL4-3-based mutants carrying the *vpu* genes, SOE-PCR was used to add SacII and NcoI sites, which allowed cloning into the pBR_NL4-3 (*nef* STOP) (*env* STOP) IRES eGFP construct[27]. The integrity of all PCR-derived inserts was verified by sequence analysis and transfections were performed using the calcium phosphate precipitation method.

**Viral infectivity**. To determine the infectious virus yield, TZM-bl cells were seeded in 96-F-well plates at a density of 6000 cells/well and infected with virus stocks containing 10 ng of p27 capsid antigen or with serial dilutions of virus stocks produced by transiently transfected HEK293T cells. Two days post-infection, viral infectivity was measured using a galactosidase screen kit (ThermoFisher) as recommended by the manufacturer. β-Galactosidase activities were quantified as relative light units (RLU) per second with an Orion Microplate luminometer (Berthold).

**SERINC5 counteraction**. To measure Nef-mediated antagonism of SERINC5, HEK293T cells were transfected with 3 μg of HIV-1 (NL4-3 based) proviral constructs containing various *nef* alleles and 2.5 μg pBJ6 AGM SERINC5-HA[65] expression plasmid or pBJ6 empty vector[66]. Two days post-transfection supernatants were harvested and analyzed for infectious HIV-1 yield by TZM-bl infection assay.

**ELISA**. SIVagm p27 core antigen was quantified using the SIV-1 p27 antigen capture assay kit (Zeptometrix Corp., Buffalo, NY) according to the manufacturer's instructions. HIV-1 p24 antigen levels were determined using an in-house ELISA. Briefly, 96-well MaxiSorp™ microplates were coated with an anti-HIV-1 p24 (Abcam) and bound capsid proteins were detected using a polyclonal rabbit antiserum against p24 antigen (1:1000, Eurogenetec) and an anti-rabbit HRP-coupled antibody (1:2000, Dianova) with a Thermo Max microplate reader (Molecular devices).

**NF-κB dual luciferase assay**. HEK293T cells were cotransfected with a firefly luciferase reporter construct under the control of three NF-κB binding sites[55], a *Gaussia* luciferase construct[55] for normalization, and expression vectors for a constitutively active mutant of IKKβ[56] and the functional *vpu* alleles as described[15]. Two days post-transfection, *Gaussia* luciferase activity in the supernatant was measured using the *Gaussia*-Juice Kit (p.j.k.) and the cells were used to measure firefly luciferase activity using the Luciferase Assay from Promega with an Orion Microplate luminometer (Berthold).

**Immunoblotting**. Transfected HEK293T cells were lysed in 500 μl radio-immunoprecipitation assay (RIPA) buffer (1% Triton X-100 (v/v), 0.15 M NaCl, 50 mM Tris (pH 7.4), 5 mM EDTA, 1 mM Phenylmethylsulfonyl fluoride in distilled water) or Co-IP buffer (150 mM NaCl, 50 mM HEPES, 5 mM EDTA, 0.1% NP40, 500 μM Na$_3$VO$_4$, 500 μM NaF, pH 7.5). Proteins were separated on a 4–12% NuPAGE Novex Bis-Tris precast gel (Invitrogen) and transferred onto a nitro-cellulose membrane (Millipore, Schwalbach) or an Immobilon-FL PVDF membrane (Merck Millipore) using a Trans-Blot SD Semi-Dry electrophoretic Transfer Cell (BioRad, Munich). The transferred membranes were blocked for 1 h with 10% milk solution and stained with anti-AU1, anti-GFP, and anti-GAPDH antibodies. Secondary anti-mouse or anti-rabbit IRDye Odyssey antibodies (1:20,000, LI-COR) were used and the fluorescent signal detected using a LI-COR Odyssey scanner and the software LI-COR Image Studio Lite Version 3.1. Viral Env expression was detected using serum from an AGM infected with SIVagmsab92018ivTF probed with horseradish peroxidase-conjugated goat anti-human immunoglobulin G (γ-chain-specific) secondary antibody (Sigma). Antibody complexes were detected by DAB (3,3′-diaminodbenzidine; Vector Labs).

To assess the establishment of a humoral immune response directed against SIVagm proteins in the AGMs, membranes were cut into 3 mm wide strips and saturated with phosphate-buffered saline (PBS) containing 5% non-fat dry milk at RT for 1 h. Plasma was diluted 1:100 in PBS supplemented with 1% non-fat dry milk, added to the strips, and incubated for 2 h. Bound antibodies were reacted with 1:1000 diluted alkaline phosphatase-conjugated goat anti-human IgG (Jackson ImmunoResearch, Europe Ltd) for 1 h followed by final color development in the dark with a one-component BCIP/NBT substrate (Moss Inc.). Initially, serum from an AGM naturally infected with SIV served as positive control, but because of weak reaction was exchanged with plasma from one of the experimentally with wild-type infected AGMs obtained 54 weeks after experimental infection.

**Viral replication in AGM PBMCs**. PBMCs from SIV-negative AGMs were stimulated with 10 μg/ml PHA for 2 days followed by overnight incubation in IL-2 medium. Activated PBMCs ($5 \times 10^6$) were infected with virus stocks containing 4 ng of p27 capsid antigen of the SIVagmSab92018ivTF clone and its derivatives. Subsequently, cells were washed extensively to remove cell-free virus and maintained in IL-2 medium. Virus production in culture supernatants was monitored at regular intervals by SIV p27 antigen capture assay (Zeptometrix).

**Viral RNA quantification**. Viral RNA was extracted from 200 μl plasma using the MagAttract Virus Mini kit (Qiagen, Hilden, Germany) and the M48 robotic system (Qiagen, Hilden, Germany). For quantification, 5 μl of the eluates were reverse transcribed and amplified using the one-step QuantiTect Probe RT-PCR Kit (Qiagen) and the 7500 Real Time PCR system (Applied Biosystems) according to the manufacturer's description. The reaction mixture contained 10 μM of each oligonucleotide: LTR forward (5′-CTGGGTGTTCTCTGGTAAG-3′), LTR- reverse (5′-CAAGACTTTATTGAGGCAAT-3′), and probe (6-carboxyfluorescein-CGAACACCCAGGCTCAAGCTGG-6-carboxytetramethyl-rhodamine)[64]. Reverse transcription was performed for 30 min at 50 °C, followed by a denaturing step at 95 °C for 10 s. Amplification was performed for 45 cycles: 15 s at 95 °C, 45 s at 55 °C, and 34 s at 72 °C. For calculation of absolute viral RNA copy numbers, a serially diluted standard RNA was reverse transcribed and amplified in parallel. Cloning and in vitro transcription of the standard RNA was done as described[67].

**RNA Purification**. Total RNA extraction was performed using the Paxgene RNA purification kit (Qiagen, Hilden, Germany). Briefly, 2.5 ml of whole blood was collected in Paxgene blood RNA tubes and total RNA extraction was performed using the Paxgene RNA purification kit according to manufacturer's specifications; on-column DNAse digestion was also performed to remove Genomic DNA. RNA integrity of the extracted RNA was assessed by Agilent Bioanalyzer (Agilent Technologies, Santa Clara, CA, USA) capillary electrophoresis on a Bioanlayzer RNA nanochip.

**Microarray hybridization**. For each individual sample, cDNA synthesis and amplification was performed using the NuGEN Ovation Whole blood solution (NuGEN, San Carlos, CA, USA). Briefly, 50 ng of total RNA was used for cDNA synthesis followed by whole transcriptome amplification by NuGEN's Ribo-SPIA® technology, this kit effectively mitigate the effects of globin transcripts in whole blood derived RNA. The Ribo-SPIA® technology uses DNA/RNA chimeric primers to amplify cDNA isothermally maintaining the stoichiometry of the input RNA. The amplified single stranded DNA was purified using the AMpure XP beads (Beckman, Indianapolis, IN, USA). Qualitative and quantitative analyses were performed on the Bioanalyzer and NanoDrop respectively to assess the size distribution of the amplified DNA and quantity. Approximately 4 μg of the amplified DNA was used for biotinylation and fragmentation using the NuGEN Ovation Encore Biotin Module (Nugen, San Carlos, CA, USA). All samples were hybridized to Affymetrix GeneChip® Rhesus Macaque Genome Arrays (Affymetrix, Santa Clara, CA, USA), which contains over 52,000 individual probe sets that assay over 47,000 transcripts. The probe arrays were washed, stained, and scanned as described in the Affymetrix GeneChip® Expression Analysis Technical Manual. CEL files were extracted from the raw scanned images using the Affymetrix GeneChip® command console Software. Quality control metrics were monitored on the Affymetrix Expression console software; discordant arrays were excluded from further downstream analyses.

**Microarray transcript differential expression analyses**. Transcript expression values were modeled and normalized using the Robust Multi-array average (RMA) method, and $\log_2$ transformed (AffyPLM, R Bioconductor). Transcripts differentially expressed between animals infected with different viral constructs were identified by a moderated Student's $t$-test (limma, R Bioconductor). Only one transcript, MmugDNA.39120.1.S1_s_at (LOC100293771), was differentially expressed at $p \leq 0.05$ with Benjamini-Hochberg multiple testing correction.

**Gene set enrichment analysis**. To identify pathways differentially modulated by the different viral constructs, gene set enrichment analysis[39] was performed as follows: For each contrast, transcripts were ranked by differential expression using the Signal2Noise metric. The "Chronic SIV immune activation" gene set was generated from a microarray analysis of SIV infection of sooty mangabeys and rhesus macaques[38] and is described in detail by Rotger and colleagues[40]. The "inflammation and immune activation" set is based on Micci and colleagues[41]. In addition to these custom gene sets, 19 immunity-related gene sets from the KEGG collection (#4062, 4610, 4611, 4612, 4620, 4621, 4622, 4623, 4640, 4650, 4657, 4658, 4659, 4660, 4662, 4664, 4666, 4670, and 4672) were obtained from the Molecular Signatures Database (http://software.broadinstitute.org/gsea/msigdb/ collections. jsp.). GSEA was performed using the desktop module available from the Broad Institute (www.broadinstitute.org/gsea/). GSEA was performed on the ranked transcript lists using 1000 phenotype permutations, collapse of duplicates to Max_probe, and random seeding.

**Fluorescent in situ hybridization**. The probes were prepared as follows. SIVenv and IL-15 mRNA were RT-PCR amplified and cDNA were cloned, using the Clo-neJET PCR Cloning Kit (Thermo Fisher Scientific) as recommended by the manufacturer. The vectors were digested and in vitro transcribed using T7 RNA polymerase (Ambion) to make Alexa Fluor 488 [Life Technologies] single-strand RNA probes. The FISH assay combined with immunofluorescent staining was performed as follows: Cryosections were rehydrated in PBS for 15 min and then permeabilized by incubating in 0.5% (v/v) Triton X-100 in PBS for 20 min at RT. The slides were placed in container filled with 200 ml of 10 mM sodium citrate

buffer (pH 6.0) and the RNA unmasked by putting the container in a microwave set at 700 W, before heating for 2.5 min or until first signs of boiling. This step was repeated seven times. The slides were transferred to 2 × SSC and subsequently incubated in formamide-SSC solution for at least 4 h. The probe was mounted using glass chambers. Prehybridization was performed by incubating the slides with the mounted probe for 1–2 h at 37 °C. Cellular RNA and RNA probes were simultaneously denatured by incubating the slides with a mounted probe on a heating block for 5 min at 80 °C, followed by hybridization in humid dark chambers for 1 day at 37 °C. Sections were washed three times at high stringency in 0.1 × SSC at 60 °C and three times in 2 × SSC buffer. Finally they were incubated in 0.05 μg/ml DAPI in SSC/Tween for 10 min, rinsed briefly in 2 × SSC and mounted. As negative controls, we used a RNAse degraded probe, as well as LNs from uninfected animals. Images were acquired on a Confocal Laser Scanning Microscope Leica TCS SP8, running LAS AF 3 (Leica Application Suite Advanced Fluorescence). Individual optical slices were collected at 1024 × 1024 pixel resolution. Image J software was used to assign colors to the channels collected.

**Immunohistochemical detection of IL-18 production**. Frozen sections of 10 μm were stained using a cross-reactive unconjugated primary antibody anti-human IL-18 (Atlas antibodies) followed by the appropriate secondary antibody conjugated to Alexa 568 (red). Prior to staining, slides were fixed in ice-cold methanol, allowed to rest for 10 min then washed 3 times with PBS. Slides were permeabilized with Tween20 and blocked with serum to avoid unspecific binding of the antibody. Confocal microscopy acquisition was performed with the Cellvoyager CV-1000 confocal microscope (Yokogawa, Japan) and images were analyzed using Image J software.

**Computer programs and data analyses**. For the analysis of nucleotide and amino acid sequences following programs were used: (i) Sequence reverse complementor (http://bioinformatics.org/ sms/ rev_comp.html)[68]; (ii) MultiAlin V5.4.1 (http://prodes.toulouse.inra.fr/multalin/)[69]; (iii) DNA/amino acid program Expasy-tool (http://www.expasy.org/tools/dna.html)[70]; (iv) Gene construction Kit V2.0 program (http://www.textco.com/gene-construction-kit.php); (v) Sequence analysis program Chroma 1.62 (http://en.bio-soft.net/format/chroma.html); and (vi) Phylogenetic tree construction by the Bayesian method using the general reversible (GTR) model of evolution.

**Quantification and statistical analysis**. Statistical analyses were performed with GraphPad PRISM (GraphPad Software) and Microsoft Excel. $p$-values were calculated using the two-tailed unpaired Student's-$t$-test. Unless otherwise stated, all experiments were performed three times and the data are shown as mean ± SEM. Significant differences are indicated as: *$p < 0.05$; **$p < 0.01$; ***$p < 0.001$.

**Data availability**. Microarray results have been deposited in the Gene Expression Omnibus database; the accession number is GSE103072. Sequences derived from AGM infected with WT or chimeric SIVagm constructs have been submitted to GenBank and are available under accession numbers MF774879 to MF775120. The authors declare that all other data supporting the findings of this study are available within the article and its Supplementary Information files, or are available from the authors upon request.

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

## Acknowledgements

We thank K. Regensburger, M. Mayer, R. Linsenmeyer, S. Engelhart, D. Krnavek, S. Heine, and J. Hampe for technical assistance, S. Sopper, K. Töpfer, and T. Schultheiss for support with flow cytometry and Nirav Patel and Greg Tharp of the Yerkes NHP Genomics Core for microarray analysis. This work was supported by NIH grants to B.H.H. (R37 AI50529, R01 AI 120810, R01 AI 114266), G.S. (R37 AI66998), and YYY (R01 AI 120860), and the BEAT-HIV Delaney Consortium (UM1 AI 126620), DFG CRC 1279 and SPP 1923, an Advanced ERC investigator grant to F.K., and a grant from Sidaction to M.M.-T. N.H. was supported by VRI (Créteil, France) and T.G.-T. by the Pasteur-Paris University PhD program. We thank the IDMIT center for access to the stellar Imaging platform and acknowledge the Imagopole France–BioImaging infrastructure, supported by the French ANR (10-INSB-04-01, Investments for the Future), for advice and access to the CV1000 system. S.J. and E.L. were part of IGradU Ulm and S.J. was supported by the DFG RTG CEMMA.

## Author contributions

F.K., C.S.-H., B.H.H., M.M.-T., G.S. and S.E.B. designed the experiments. S.J., E.H.P., C. W.G., E.L., C.M.S., N.F.P., U.S., B.N., K.M.R., D.F., J.M.B., C.A., N.H., T.G.-T. and D.H. conducted the experiments. F.K., C.S.-H., B.H.H., M.M.-T., G.H.L. and D.S. analyzed the data. F.K. and B.H.H wrote the manuscript.

## Additional information

**Competing interests:** The authors declare no competing interests.

