## [Peer Review File(PDF 294 kb) · Nature Communications]

Reviewers' comments:

Reviewer #1 (Remarks to the Author):

Comments to authors

In this manuscript, Joas et al., investigate the determinants in HIV, SIV and their primate hosts responsible for the different outcomes of these lentiviral infections. Whereas SIVs that effectively replicate in sooty mangabeys and African green monkeys cause viremia with no pathogenesis, HIV causes chronic inflammation and depletion of CD4+ T cell subsets. SIVs normally use their Nef proteins to counteract restriction by Tetherin/BST2 and down-modulate CD3 to prevent chronic T cell activation – although certain SIVs use Vpu rather than Nef to antagonize BST2. Unlike these non-pathogenic SIVs, HIV Nef proteins do not down-modulate CD3, rendering cells susceptible to chronic activation. However, this is somewhat reversed by using Vpu to block NF- κ B. Vpu is also the protein of choice for most HIVs to antagonize BST2. Therefore, it seems that the Vpu and Nef proteins of the primate lentiviruses are functionally linked, a fact that is further demonstrated in the current manuscript.

The authors of this study investigated whether the differences in Vpu and Nef between pathogenic and non-pathogenic lentiviruses are the key determinants to progress to AIDS. For this, they generated recombinant SIVagm molecular clones containing SIVgsnVpu (which retains anti-BST2 and anti-NF- κ B activity) and/or HIV1 NA7 Nef, which lacks the ability to down-modulate CD3. Although the molecular clones were infectious in vitro, some of the recombinants were not infectious in vivo (i.e. GU). Most of the recombinant clones caused immune activation in infected animals, particularly GU1N, although this virus failed at accelerating CD4 T cell depletion and immunodeficiency. The authors of this study concluded that host factors rather than viral factors of pathogenicity are the key determinants for chronic infection and AIDS, since determinants of pathogenicity in Vpu or Nef did not supersede host protective factors in species that are naturally infected with SIVs. These findings are novel and bring light into the contribution of viral proteins and host factors in the progression to AIDS. The paper is well organized, the experimental approach is well designed, appropriate statistics are included and the results obtained support the authors' conclusion. The manuscript is clearly written for the most part, although the sections describing figures 2 and 3 are heavy at times. I would encourage the authors to digest that information a bit more. Besides that, I have few minor questions that I would like the authors to address.

Minor.

SIV recombinant clone. The authors introduce SIVgsn71Vpu instead of HIV-1 Vpu reasoning that the ability of lentiviral Vpu proteins to counteract BST2 is species-specific. They further demonstrated that despite of their distinct origin, both HIV and SIVgsn Vpu proteins similarly down-regulate other cell surface molecules and that they equally block NF- κ B activation. In the case of the HIV-1 Nef, they selected the Nef protein of HIV-1 NA7 (why was the Nef protein of this isolate selected?). The authors clearly showed the differences in behavior between SIVagm and NA7 Nef proteins. However, the NA7 Nef protein retains the ability to counteract restriction by non-human primate BST2 (Jia and Serra-Moreno et al.,

2009). It does particularly well against sooty mangabey BST2, which is highly similar to agm BST2. The authors nevertheless, show that in their hands NA7 Nef does not counteract restriction by agm BST2 (Fig. 1f). These controversial results should be discussed.

Figure 1c. The assays to evaluate the functionality and resemblance between SIVgsnVpu and HIV Vpu are clear, however, it is not clear to this reviewer which BST2 protein (human or agm) was tested for this panel. As it stands, it seems only human versions of these proteins (CD4, NTBA, CD1d, and BST2) were tested. If this is the case, how do the authors explain that SIVgsn71Vpu counteracts human BST2? According to a previous paper from this group (Sauter et al., 2009) this molecular clone is inefficient at down-regulating hBST2.

Figure 1h. The molecular clones propagate in vitro, but even the GU1N replicates at a much lower rate than wild type SIV agm. This may be why at peak of viremia there is > 1-log difference between wild type and GU1N. Is not it possible that this lower level of replication can affect infectivity in vivo, in a way that the virus fails at causing AIDS?

Figure 1j. Why would the SIVagm 1N fail at downregulating MHC-I whereas the SIVagm GU1N, which encodes the same Nef variant, is perfectly capable of down-regulating this surface marker?

Figure 3c. The authors claim that all viremic viruses showed rapid decrease in circulating CCR5-CXCR3-CD4 T cells, however, this effect is not entirely true for GU infected animals. This needs to be rephrased.

Figure 4c. The authors compare how mutations accumulated in Vpu affect the functionality of the protein by checking its anti-BST2 activity. They conclude that all Vpu variants similarly counteract restriction by AGM BST2. However, some variants (14633, 14634 for instance) retain almost 100% infectivity at the highest levels of BST2 expression, and thus perform better than wild type SIVgsn. As the authors claim, when looking at virion release levels (p24 levels) from AGM BST2 expressing cells, there is not much of a difference. Therefore, although the Vpu variants seem to retain similar properties to counteract BST2 and facilitate virion release, it seems that these Vpu proteins might increase the infectivity of the virions. This needs to be discussed

Figure 6d. The authors claim that genes from KEEGG Immunity collection were enriched only in GU1N infected animals. However, there are several instances in which there is enrichment in the GU-infected animals. This needs to be addressed

Page 16 line 9. The authors should include additional references for the gain of anti-BST2 activity by Env (SIV tan and SIVmac239 Δ nef, Gupta et al., 2009; and Serra-Moreno et al., 2011)

Reviewer #2 (Remarks to the Author):

In this report by Joas and colleagues, the authors report the construction and characterization of SIVagm derived viruses that recapitulate putative HIV-1 pathogenicity determinants; namely a functional Vpu protein that mediates Tetherin downregulation (and other functions) and a Nef allele that fails to downregulate CD3, or both. Overall, the study is clever and addresses important questions about how HIV-1 may have become an epidemic inducing immunodeficiency virus.

Positives include;

- 1) the well executed and presented experiments, both in vitro and in vivo
- 2) the clarity of the report, and the reporting of several novel and important findings.
- 3) the accidental infection by animals in adjacent cages is clearly described and the conclusions based on these data are justified and the overall conclusions are not negatively impacted
- 4) the description of how the authors had to mutate an overly strong Kozak site upstream of the Vpu ORF to enable sufficient Env expression was adequately justified and explained. These details do not detract from the report. In my view, they strengthen it.

Important findings include;

- 1) the highly efficient in vivo replication of the GU1N virus
- 2) the surprising stability of the HIV and gsn viral gene products over long term infection in AGMs
- 3) the lack of immunodeficiency disease in infected animals despite an increase in inflammation
- 4) Most important: the finding that there is a strong functional linkage between Vpu and Nef, with specific functions of these proteins being critical to the linkage. By combining these gene products, with well characterized functions, onto a viral backbone allowed the authors to assess their importance, singly and in concert. This is an elegant and clever approach to assessing the importance of viral gene products on viral replication in vitro and in vivo.

Negatives include;

- 1) The title, though accurate, is underwhelming and undersells the story.
- 2) The pattern of IL-18 staining suggested in Fig 7 is apparent but holds little weight in the absence of any statistical analysis to compare animals infected with the different viruses. These data would be strengthened by staining for additional markers of inflammation or a more thorough analysis of the data presented.
- 3) Is it appropriate to use microarray chips with rhesus sequences to monitor expression of AGM gene expression? Presumably rhesus is the closest relative to AGM with available reagents for microarray? The authors should explain or cite reports justifying this. Otherwise, the microarray data is robust and well presented.
- 4) It's shown that the gsn Vpu downregulates AGM Tetherin, but other activities of this Vpu allele are tested against human proteins. Is it possible this Vpu is not active against the AGM versions of these proteins? HIV-1 Nef and Vpu act coordinately to modulate expression

of multiple human proteins, including CD4 and CD1d. If this Vpu allele is not fully active against AGM proteins, it could explain the lack of immunodeficiency (in addition to host factors). This should be clarified.

5) This report describes a large number of expertly conducted experiments but as it is written, it is difficult to understand how all these data contribute to the conclusion that host factors, rather than virus factors, determine the lack of pathogenesis in natural hosts. Specifically, I see the second to last paragraph of the Introduction, beginning on page 3, line 19, as critical to the story of how the acquisition of a vpu gene and loss of Nef-mediated CD3 downregulation may have been critical determinants of HIV-1 pathogenesis (which is the crux of the hypothesis being tested). Yet this paragraph is confusing as written. Specifically, readers not versed in the importance of Nef mediated CD3 downregulation and how this differs between HIV-1 and its immediate ancestors and most/all other SIVs, will likely not follow this text as written. This paragraph should be re-written or rearranged to provide clarity.

Rebuttal (reviewers' comments are in *italic* letters):

Reviewer #1 noted that our findings “*are novel and bring light into the contribution of viral proteins and host factors in the progression to AIDS*”. He/she felt that “*the paper is well organized, the experimental approach is well designed, appropriate statistics are included and the results obtained support the authors' conclusion*” and raised the following “*Specific minor points*”:

1. *In the case of the HIV-1 Nef, they selected the Nef protein of HIV-1 NA7 (why was the Nef protein of this isolate selected?). The authors clearly showed the differences in behavior between SIVagm and NA7 Nef proteins. However, the NA7 Nef protein retains the ability to counteract restriction by non-human primate BST2 (Jia and Serra-Moreno et al., 2009). It does particularly well against sooty mangabey BST2, which is highly similar to agm BST2. The authors nevertheless, show that in their hands NA7 Nef does not counteract restriction by agm BST2 (Fig. 1f). These controversial results should be discussed.*

To address these issues, we now provide a rationale for selection of the NA7 Nef (pg 5, lines 15-16). We also mention the previous findings of Jia and colleagues (pg 5, lines 21-23). We do not believe that their results are in disagreement with ours. Jia and colleagues show that the HIV-1 NA7 Nef fails to counteract human tetherin, but has some activity against sooty mangabey tetherin. However, in this same study rhesus tetherin was hardly counteracted by HIV-1 NA7 Nef, although it is also closely related to AGM tetherin. Thus, the NA7 Nef seems to show modest activity against some, but not all, non-human versions of tetherin.

2. *Figure 1c. The assays to evaluate the functionality and resemblance between SIVgsnVpu and HIV Vpu are clear, however, it is not clear to this reviewer which BST2 protein (human or agm) was tested for this panel. As it stands, it seems only human versions of these proteins (CD4, NTBA, CD1d, and BST2) were tested. If this is the case, how do the authors explain that SIVgsn71Vpu counteracts human BST2? According to a previous paper from this group (Sauter et al., 2009) this molecular clone is inefficient at down-regulating hBST2.*

We apologize for the lack of clarity. As now specified in the revised manuscript (pg 5, lines 7-14), tetherin was derived from AGM (which explains the activity of the SIVgsn Vpu), while the other proteins were of human origin. Vpu and Nef are known to counteract restriction factors in a species-specific manner, while their effects on immune receptors are largely species independent. We now justify our rationale and describe the conservation of potential Vpu and Nef target sites in the revised manuscript.

3. *Figure 1h. The molecular clones propagate in vitro, but even the GUIN replicates at a much lower rate than wild type SIVagm. This may be why at peak of viremia there is > 1-log difference between wild type and GUIN. Is not it possible that this lower level of replication can affect infectivity in vivo, in a way that the virus fails at causing AIDS?*

Yes, the relative attenuation of the GUIN SIVagm construct early during acute infection might have contributed to the lack of disease progression. We now mention this in the discussion section (pg 16, lines 7-10). We plan to examine the properties of the SIVagm GUIN strains obtained at necropsy in a follow-up study. However, these experiments will be very time consuming and are beyond the scope of the present study. Importantly, both viral replication and immune activation are thought to drive disease progression during chronic infection and the viral loads in WT and GUIN infected AGM were essentially identical for the last 3 years of follow-up.

4. *Figure 1j. Why would the SIVagm 1N fail at downregulating MHC-I whereas the SIVagm GUIN, which encodes the same Nef variant, is perfectly capable of down-regulating this surface marker?*

The SIVagm 1N construct downmodulated MHC-I by about 30%. As mentioned in the paper (pg 6, line 25 to pg 7, line 2), Vpu also targets HLA-C and downmodulates MHC-I by suppressing NF- κ B activity. Thus, in the case of GUIN, Vpu and Nef most likely cooperate to counteract MHC-I.

5. *Figure 3c. The authors claim that all viremic viruses showed rapid decrease in circulating CCR5-CXCR3-CD4 T cells, however, this effect is not entirely true for GU infected animals. This needs to be rephrased.*

We feel that our statement is correct since GU infected animals were non-viremic. However, we have added a sentence for clarification (pg 9, lines 7/8).

6. *Figure 4c. The authors compare how mutations accumulated in Vpu affect the functionality of the protein by checking its anti-BST2 activity. They conclude that all Vpu variants similarly counteract restriction by AGM BST2. However, some variants (14633, 14634 for instance) retain almost 100% infectivity at the highest levels of BST2 expression, and thus perform better than wild type SIVgsn. As the authors claim, when looking at virion release levels (p24 levels) from AGM BST2 expressing cells, there is not much of a difference. Therefore, although the Vpu variants seem to retain similar properties to counteract BST2 and facilitate virion release, it seems that these Vpu proteins might increase the infectivity of the virions. This needs to be discussed.*

The reviewer is correct that we observed a 25% reduction in infectious virus yield for wild-type and no reduction for 14633 and 14634 Vpus (Fig. 4C). We now mention this in the revised manuscript (pg 11, lines 15-17). However, these differences are rather subtle and altogether we did not observe statistical significant differences in the effect of these Vpu proteins on virion infectivity.

7. *Figure 6d. The authors claim that genes from KEEGG Immunity collection were enriched only in GUIN infected animals. However, there are several instances in which there is enrichment in the GU-infected animals. This needs to be addressed.*

We thank the reviewer for raising this interesting point. To clarify this, we analyzed the presence of SIV+ cells in tissues from all AGMS. Our new data show that despite undetectable viral load and lack of virus re-isolation SIV infected cells are detectable in lymphoid tissues of the GU group (new Fig. 7 and Supplementary Fig. 6). Thus, as now mentioned in the revised manuscript (pg 13, lines 10-16; pg 13, line 20 to pg 14, line 2) low level viral persistence in lymphoid tissues might stimulate expression of some immune genes in the GU group. Further examination of lymphoid tissues is planned, but the experiments are very time-consuming and are beyond the scope of the present manuscript, which already contains a large amount of data.

8. *Page 16 line 9. The authors should include additional references for the gain of anti-BST2 activity by Env (SIV tan and SIVmac239 Δ nef, Gupta et al., 2009; and Serra-Moreno et al., 2011).*

These references have been added (pg 17, lines 6-8).

Reviewer #2 wrote that *“the study is clever and addresses important questions about how HIV-1 may have become an epidemic inducing immunodeficiency virus”*. He/she felt that positive aspects are (amongst others) *“well executed and presented experiments”* as well as *“the reporting of several novel and important findings”*. Reviewer 2 noted that our study describes several important findings on viral accessory gene function. This reviewer raised the following issues:

1. *The title, though accurate, is underwhelming and undersells the story.*

We changed the title to may it more appealing to a broad readership.

2. *The pattern of IL-18 staining suggested in Fig 7 is apparent but holds little weight in the absence of any statistical analysis to compare animals infected with the different viruses. These data would be strengthened by staining for additional markers of inflammation or a more thorough analysis of the data presented.*

To address this point, we tried to perform quantitative analyses and examined IL-18 expression in another tissue (ileum). Unfortunately, signal intensities were highly variable in different tissue sections and not significantly different between the groups. As now discussed in the revised manuscript (pg 14, lines 11/12), further studies in a larger number of animals are required to fully define potential differences in inflammation markers.

3. *Is it appropriate to use microarray chips with rhesus sequences to monitor expression of AGM gene expression? Presumably rhesus is the closest relative to AGM with available reagents for microarray? The authors should explain or cite reports justifying this. Otherwise, the microarray data is robust and well presented.*

The reviewer is correct that the data were generated using a commercially available microarray designed for a closely related primate species. We now discuss the suitability of this array for studies in AGMs in the revised manuscript (pg 12, lines 17-22).

4. *It's shown that the gsn Vpu downregulates AGM Tetherin, but other activities of this Vpu allele are tested against human proteins. Is it possible this Vpu is not active against the AGM versions of these proteins? HIV-1 Nef and Vpu act coordinately to modulate expression of multiple human proteins, including CD4 and CD1d. If this Vpu allele is not fully active against AGM proteins, it could explain the lack of immunodeficiency (in addition to host factors). This should be clarified.*

The reviewer is correct that we cannot formally exclude the possibility that the SIV_{gsn} Vpu is not fully active against some AGM proteins. However, as mentioned in the revised manuscript (pg. 5, lines 7-14), the effects of accessory proteins on these immune receptors are usually species-independent. Moreover, the SIV_{gsn} Vpu downmodulated human CD4 and CD1d as effectively as the HIV-1 Vpu (Fig. 1c). The high viral loads observed in GU1N infected animals suggest that the SIV_{gsn} Vpu promotes efficient immune evasion in AGMs and the effects on some receptors were verified in primary AGM cells (Fig. 1j).

5. *This report describes a large number of expertly conducted experiments but as it is written, it is difficult to understand how all these data contribute to the conclusion that host factors, rather than virus factors, determine the lack of pathogenesis in natural hosts. Specifically, I see the second to last paragraph of the Introduction, beginning on page 3, line 19, as critical to the story of how the acquisition of a vpu gene and loss of Nef-mediated CD3 downregulation may have been critical determinants of HIV-1 pathogenesis (which is the crux of the hypothesis being tested). Yet this paragraph is confusing as written. Specifically, readers not versed in the importance of Nef mediated CD3 downregulation and how this differs between HIV-1 and its immediate ancestors and most/all other SIVs, will likely not follow this text as written. This paragraph should be re-written or rearranged to provide clarity.*

We agree with the reviewer that the rationale for our experimental design has to be explained more clearly. We have thus modified and expanded the Introduction to describe the potential role of CD3 downmodulation in viral pathogenesis in greater detail by highlighting the differences between HIV-1 and other primate lentiviruses (pg. 4, lines 1-4).

REVIEWERS' COMMENTS:

Reviewer #1 (Remarks to the Author):

The authors have satisfactorily addressed all the concerns raised in the previous round of revisions. Their explanations, edits to the text and figures have strengthened their manuscript, and I feel it should be accepted for publication.

Reviewer #2 (Remarks to the Author):

I appreciate the authors' carefully addressing the reviewer concerns and i'm comfortable that they have adequately addressed the points i raised.